# Monotone deep Boltzmann machines

**Zhili Feng**                                                          *zhilif@andrew.cmu.edu*
*Machine Learning Department*
*Carnegie Mellon University*

**Ezra Winston**                                                        *ewinston@cs.cmu.edu*
*Machine Learning Department*
*Carnegie Mellon University*

**J. Zico Kolter**                                                      *zkolter@cs.cmu.edu*
*Computer Science Department*
*Carnegie Mellon University*
*Bosch Center for AI*

**Reviewed on OpenReview:** *https://openreview.net/forum?id=SgTKk6ryPr*

## Abstract

Deep Boltzmann machines (DBMs), one of the first "deep" learning methods ever studied, are multi-layered probabilistic models governed by a pairwise energy function that describes the likelihood of all variables/nodes in the network. In practice, DBMs are often constrained, i.e., via the *restricted* Boltzmann machine (RBM) architecture (which does not permit intra-layer connections), in order to allow for more efficient inference. In this work, we revisit the generic DBM approach, and ask the question: are there other possible restrictions to their design that would enable efficient (approximate) inference? In particular, we develop a new class of restricted model, the monotone DBM, which allows for arbitrary self-connection in each layer, but restricts the *weights* in a manner that guarantees the existence and global uniqueness of a mean-field fixed point. To do this, we leverage tools from the recently-proposed monotone Deep Equilibrium model and show that a particular choice of activation results in a fixed-point iteration that gives a variational mean-field solution. While this approach is still largely conceptual, it is the first architecture that allows for efficient approximate inference in fully-general weight structures for DBMs. We apply this approach to simple deep convolutional Boltzmann architectures and demonstrate that it allows for tasks such as the joint completion and classification of images, within a single deep probabilistic setting, while avoiding the pitfalls of mean-field inference in traditional RBMs.

## 1 Introduction

This paper considers (deep) Boltzmann machines (DBMs), which are pairwise energy-based probabilistic models given by a joint distribution over variables $\mathbf{x}$ with density

$$p(\mathbf{x}) \propto \exp\left( \sum_{(i,j) \in E} x_i^\top \Phi_{ij} x_j + \sum_{i=1}^n b_i^\top x_i \right), \tag{1}$$

where each $x_{1:n}$ denotes a discrete random variable over $k_i$ possible values, represented as a one-hot encoding $x_i \in \{0,1\}^{k_i}$; $E$ denotes the set of edges in the model; $\Phi_{ij} \in \mathbb{R}^{k_i \times k_j}$ represents pairwise potentials; and $b_i \in \mathbb{R}^{k_i}$ represents unary potentials. Depending on the context, these models are typically referred to as pairwise Markov random fields (MRFs) (Koller & Friedman, 2009), or (potentially deep) Boltzmann machines (Goodfellow et al., 2016; Salakhutdinov & Hinton, 2009; Hinton, 2002). In the above setting, each $x_i$ may

Figure 1: Neural network topology of different Boltzmann machines. The general case is a complete graph (red dashed lines are a subset of edges that are in BM but not RBM). Our proposed parameterization is a form of general Boltzmann machine.

represent an observed or unobserved value, and there can be substantial structure within the variables; for instance, the collection of variables $\mathbf{x}$ may (and indeed will, in the main settings we consider in this paper) consist of several different "layers" in a joint convolutional structure, leading to the deep convolutional Boltzmann machine (Norouzi et al., 2009).

Boltzmann machines were some of the first "deep" networks ever studied Ackley et al. (1985). However, in modern deep-learning practice, general-form DBMs have largely gone unused, in favor of *restricted* Boltzmann machines (RBMs). These are DBMs that avoid any connections within a single layer of the model and thus lead themselves to more efficient block-based approximate inference methods.

In this paper, we revisit the general framework of a *generic* DBM, and ask the question: are there any other restrictions (besides avoiding intra-layer connections), that would *also* allow for efficient approximate inference methods? To answer this question, we propose a new class of general DBMs, the *monotone deep Boltzmann machine* (mDBM); unlike RBMs, these networks can have dense intra-layer connections but are parameterized in a manner that constrains the weights so as to still guarantee an efficient inference procedure. Specifically, in these networks, we show that there is a unique and globally optimal fixed point of variational mean-field inference; this contrasts with traditional probabilistic models where mean-field inference may lead to multiple different local optima. To accomplish this goal, we leverage recent work on monotone Deep Equilibirum (monDEQ) models (Winston & Kolter, 2020), and show that a particular choice of activation function leads to a fixed point iteration equivalent to (damped) parallel mean-field updates. Such fixed point iterations require the development of a new proximal operator method, for which we derive a highly efficient GPU-based implementation.

Our method also relates closely with previous works on convergent mean-field inference in Markov random fields (MRFs) (Krähenbühl & Koltun, 2013; Baqué et al., 2016; Lê-Huu & Alahari, 2021); but these approaches either require stronger conditions on the network or fail to converge to the true mean-field fixed point, and generally have only been considered on standard "single-layer" MRFs. Our approach can be viewed as a combined model parameterization and (properly damped) mean-field inference procedure, such that the resulting iteration is guaranteed to converge to a unique optimal mean-field fixed point when run in parallel over all variables.

Although the approach is still largely conceptual, we show for the first time that one can learn and perform inference in structured multi-layer Boltzmann machines which contain intra-layer connections. For example, we perform both learning and inference for a deep convolutional, multi-resolution Boltzmann machine, and apply the network to model MNIST and CIFAR-10 pixels and their classes conditioned on partially observed images. Such joint probabilistic modeling allows us to simultaneously impute missing pixels and predict the class. While these are naturally small-scale tasks, we emphasize that performing joint probabilistic inference over a complete model of this type is a relatively high-dimensional task as far as traditional mean-field inference is concerned. We compare our approach to (block structured) mean-field inference in classical RBMs, showing substantial improvement in these estimates, and also compare to alternative mean-field

inference approaches. Although in initial phases, the work hints at potential new directions for Boltzmann machines involving very different types of restrictions than what has typically been considered in deep learning.

## 2 Background and related work

This paper builds upon three main avenues of work: 1) deep equilibrium models, especially one of their convergent version, the monotone DEQ; 2) the broad topic of energy-based deep model and Boltzmann machines in particular; and 3) work on concave potentials and parallel methods for mean-field inference. We discuss each of these below.

**Equilibrium models and their provable convergence**  The DEQ model was first proposed by Bai et al. (2019). Based on the observation that a neural network $q^{t+1} = \sigma(Wq_t + Ux + b)$ with input injection $x$ usually converges to a fixed point, they modeled an effectively infinite-depth network with input injection directly via its fixed point: $q^* = \sigma(Wq^* + Ux + b)$. Its backpropagation is done through the implicit function theorem and only requires constant memory. Bai et al. (2020) also showed that the multiscale DEQ models achieve near state-of-the-art performances on many large-scale tasks. Winston & Kolter (2020) later presented a parametrization of the DEQ (denoted as monDEQ) that guarantees provable convergence to a unique fixed point, using monotone operator theory. Specifically, they parameterize $W$ in a way that $I - W \succeq mI$ (called $m$-strongly monotone) is always satisfied during training for some $m > 0$; they convert nonlinearities into proximal operators (which include ReLU, tanh, etc.), and show that using existing splitting methods like *forward-backward* and *Peaceman-Rachford* can provably find the unique fixed point. Other notable related implicit model works include Revay et al. (2020), which enforces Lipschitz constraints on DEQs; El Ghaoui et al. (2021) provides a thorough introduction to implicit models and their well-posedness. Tsuchida & Ong (2022) also solves graphical model problems using DEQs, focusing on principal component analysis.

**Markov random field (MRF) and its variants**  MRF is a form of energy-based models, which model joint probabilities of the form $p_\theta(x) = \exp(-E_\theta(x))/Z_\theta$ for an energy function $E_\theta$. A common type of MRF is the Boltzmann machine, the most successful variant of which is the restricted Boltzmann machines (RBM) (Hinton, 2002) and its deep (multi-layer) variant (Salakhutdinov & Hinton, 2009). Particularly, RBMs define $E_\theta(v, h) = -a^\top v - b^\top h - v^\top W h$, where $\theta = \{W, a, b\}$, $v$ is the set of visible variables, and $h$ is the set of latent variables. It is usually trained using the contrastive-divergence algorithm, and its inference can be done efficiently by a block mean-field approximation. However, a particular restriction of RBMs is that there can be no intra-layer connections, that is, each variable in $v$ (resp. $h$) is independent conditioned on $h$ (resp. $v$). A deep RBM allows different layers of hidden nodes, but there cannot be intra-layer connections. By contrast, our formulation allows intra-layer connections and is therefore more expressive in this respect. See Figure 1 for the network topology of RBM, deep RBM, and general BM (we also use the term general deep BM interchangeably to emphasize the existence of deep structure). Wu et al. (2016) proposed a deep parameterization of MRF, but their setting only considers a grid of hidden variables $h$, and the connections among hidden units are restricted to the neighboring nodes. Therefore, it is a special case of our parameterization (although their learning algorithm is orthogonal to ours). Numerous works also try to combine deep neural networks with conditional random fields (CRF) (Krähenbühl & Koltun, 2013; Zheng et al., 2015; Schwartz et al., 2017) These models either train a pre-determined kernel as an RNN or use neural networks for producing either inputs or parameters of their CRFs.

**Parallel and convergent mean-field**  It is well-known that mean-field updates converge locally using a coordinate ascent algorithm (Blei et al., 2017). However, local convergence is only guaranteed if the update is applied sequentially. Nonetheless, several works have proposed techniques to parallelize updates. Krähenbühl & Koltun (2013) proposed a concave-convex procedure (CCCP) to minimize the KL divergence between the true distribution and the mean-field variational family. To achieve efficient inference, they use a concave approximation to the pairwise kernel, and their fast update rule only converges if the kernel function is concave. Later, Baqué et al. (2016) derived a similar parallel damped forward iteration to ours that provably converges without the concave potential constraint. However, unlike our approach, they do not use a parameterization that ensures a global mean-field optimum, and their algorithm therefore may

not converge to the actual fixed point of the mean-field updates. This is because Baqué et al. (2016) used the $\text{prox}_f^1$ proximal operator (described below), whereas we derive the $\text{prox}_f^\alpha$ operator to guarantee global convergence when doing mean-field updates in parallel. What's more, Baqué et al. (2016) focused only on inference over prescribed potentials, and not on training the (fully parameterized) potentials as we do here. Lê-Huu & Alahari (2021) brought up a generalized Frank-Wolfe based framework for mean-field updates which include the methods proposed by Baqué et al. (2016); Krähenbühl & Koltun (2013). Their results only guarantee global convergence to a local optimal.

## 3 Monotone deep Boltzmann machines and approximate inference

In this section, we present the main technical contributions of this work. We begin by presenting a parameterization of the pairwise potential in a Boltzmann machine that guarantees the monotonicity condition. We then illustrate the connection between a (joint) mean-field inference fixed point and the fixed point of our monotone Boltzmann machine (mDBM) and discuss how deep structured networks can be implemented in this form practically; this establishes that, under the monotonicity conditions on $\mathbf{\Phi}$, there exists a unique globally-optimal mean-field fixed point. Finally, we present an efficient parallel method for computing this mean-field fixed point, again motivated by the machinery of monotone DEQs and operator splitting methods.

### 3.1 A monotone parameterization of general Boltzmann machines

In this section, we show how to parameterize our probabilistic model in a way that the pairwise potentials satisfy $I - \mathbf{\Phi} \succeq mI$, which will be used later to show the existence of a unique mean-field fixed point. Recall that $\mathbf{\Phi}$ defines the interaction between random variables in the graph. In particular, for random variables $x_i \in \mathbb{R}^{k_i}$, $x_j \in \mathbb{R}^{k_j}$, we have $\mathbf{\Phi}_{ij} \in \mathbb{R}^{k_i \times k_j}$. Additionally, since $\mathbf{\Phi}$ defines a graphical model that has no self-loop, we further require $\mathbf{\Phi}$ to be a *block hollow* matrix (that is, the $k_i \times k_i$ diagonal blocks corresponding to each variable must be zero). While both these conditions on $\mathbf{\Phi}$ are convex constraints, in practice it would be extremely difficult to project a generic set of weights onto this constraint set under an ordinary parameterization of the network.

Thus, we instead advocate for a *non-convex* parameterization of the network weights, but one which guarantees that the monotonicity condition is always satisfied, without any constraint on the weights in the parameterization. Specifically, define the block matrix

$$\boldsymbol{A} = \left[ \begin{array}{cccc} A_1 & A_2 & \cdots & A_n \end{array} \right]$$

with $A_i \in \mathbb{R}^{d \times k_i}$ matrices for each variables, and where $d$ can be some arbitrarily chosen dimension. Then let $\hat{A}_i$ be a spectrally-normalized version of $A_i$

$$\hat{A}_i = A_i \cdot \min\{\sqrt{1-m}/\|A_i\|_2, 1\} \tag{2}$$

i.e., a version of $A_i$ normalized such that its largest singular value is at most $\sqrt{1-m}$ (note that we can compute the spectral norm of $A_i$ as $\|A_i\|_2 = \|A_i^T A_i\|_2^{1/2}$, which involves computing the singular values of only a $k_i \times k_i$ matrix, and thus is very fast in practice). We define the $\hat{\boldsymbol{A}}$ matrix analogously as the block version of these normalized matrices.

Then we propose to parameterize $\mathbf{\Phi}$ as

$$\mathbf{\Phi} = \text{blkdiag}(\hat{\boldsymbol{A}}^T \hat{\boldsymbol{A}}) - \hat{\boldsymbol{A}}^T \hat{\boldsymbol{A}} \tag{3}$$

where blkdiag denotes the block-diagonal portion of the matrix along the $k_i \times k_i$ block. Put another way, this parameterizes $\mathbf{\Phi}$ as

$$\Phi_{ij} = \begin{cases} -\hat{A}_i^T \hat{A}_j & \text{if } i \neq j, \\ 0 & \text{if } i = j. \end{cases} \tag{4}$$

As the following simple theorem shows, this parameterization guarantees both hollowness of the $\mathbf{\Phi}$ matrix and monotonicity of $I - \mathbf{\Phi}$, for any value of the $\boldsymbol{A}$ matrix.

**Theorem 3.1.** *For any choice of parameters $\boldsymbol{A}$, under the parametrization equation 3 above, we have that 1) $\Phi_{ii} = 0$ for all $i = 1, \ldots, n$, and 2) $I - \boldsymbol{\Phi} \succeq mI$.*

*Proof.* Block hollowness of the matrix follows immediately from construction. To establish monotonicity, note that

$$
\begin{aligned}
I - \boldsymbol{\Phi} \succeq mI &\iff I + \hat{\boldsymbol{A}}^T \hat{\boldsymbol{A}} - \text{blkdiag}(\hat{\boldsymbol{A}}^T \hat{\boldsymbol{A}}) \succeq mI \\
&\impliedby I - \text{blkdiag}(\hat{\boldsymbol{A}}^T \hat{\boldsymbol{A}}) \succeq mI \iff I - \hat{A}_i^T \hat{A}_i \succeq mI, \ \forall i \\
&\iff \|\hat{A}_i\|_2 \le \sqrt{1-m}, \ \forall i.
\end{aligned}
\tag{5}
$$

This last property always holds by construction of $\hat{A}_i$. $\qquad\square$

### 3.2 Mean-field inference as a monotone DEQ

In this section, we formally present how to formulate the mean-field inference as a DEQ update. Recall from before that we are modeling a distribution of the form Equation (1). We are interested in approximating the conditional distribution $p(\mathbf{x}_h|\mathbf{x}_o)$, where $o$ and $h$ denote the observed and hidden variables respectively, with a factored distribution $q(\mathbf{x}_h) = \prod_{i \in h} q_i(x_i)$. Here, the standard mean-field updates (which minimize the KL divergence between $q(\mathbf{x}_h)$ and $p(\mathbf{x}_h|\mathbf{x}_o)$ over the single distribution $q_i(x_i)$) are given by the following equation,

$$
q_i(x_i) := \text{softmax} \left( \sum_{j:(i,j) \in E} \Phi_{ij} q_j(x_j) + b_i \right)
$$

where overloading notation slightly, we let $q_j(x_j)$ denote a one-hot encoding of the observed value for any $j \in o$ (see e.g., Koller & Friedman (2009) for a full derivation).

The essence of the above updates is a characterization of the joint fixed point to mean-field inference. For simplicity of notation, defining

$$
\boldsymbol{q} = \begin{bmatrix} q_1(x_1) & q_2(x_2) & \ldots \end{bmatrix}^T.
$$

We see that $\boldsymbol{q}_h$ is a joint fixed point of all the mean-field updates if and only if

$$
\boldsymbol{q}_h = \text{softmax} \left( \boldsymbol{\Phi}_{hh} \boldsymbol{q}_h + \boldsymbol{\Phi}_{ho} \mathbf{x}_o + \boldsymbol{b}_h \right)
\tag{6}
$$

where $\mathbf{x}_o$ analogously denotes the stacked one-hot encoding of the observed variables.

We briefly recall the monotone DEQ framework of Winston & Kolter (2020). Given input vector $\mathbf{x}$, a monotone DEQ computes the fixed point $\boldsymbol{z}^\star(\mathbf{x})$ that satisfies the equilibrium equation $\boldsymbol{z}^\star(\mathbf{x}) = \sigma(W\boldsymbol{z}^\star(\mathbf{x}) + U\mathbf{x} + b)$. Then if: 1) $\sigma$ is given by a proximal operator[1] $\sigma(x) = \text{prox}_f^1(x)$ for some convex closed proper (CCP) $f$, and 2) if we have the monotonicity condition $I - W \succeq mI$ (in the positive semidefinite sense) for some $m > 0$, then for any $\mathbf{x}$ there exists a unique fixed point $\boldsymbol{z}^\star(\mathbf{x})$, which can be computed through standard operator splitting methods, such as forward-backward splitting.

We now state our main claim of this subsection, that under certain conditions the mean-field fixed point can be viewed as the fixed point of an analogous DEQ. This is formalized in the following proposition.

**Proposition 3.1.** *Suppose that the pairwise kernel $\boldsymbol{\Phi}$ satisfies $I - \boldsymbol{\Phi} \succeq mI$ [2] for $m > 0$. Then the mean-field fixed point*

$$
\boldsymbol{q}_h = \text{softmax} \left( \boldsymbol{\Phi}_{hh} \boldsymbol{q}_h + \boldsymbol{\Phi}_{ho} \mathbf{x}_o + \boldsymbol{b}_h \right)
\tag{7}
$$

*corresponds to the fixed point of a monotone DEQ model. Specifically, this implies that for any $\mathbf{x}_o$, there exists a unique, globally-optimal fixed point of the mean-field distribution $\boldsymbol{q}_h$.*

---

[1] A proximal operator is defined by $\text{prox}_f^\alpha(x) = \arg\min_z \frac{1}{2}\|x - z\|^2 + \alpha f(z)$.

[2] Technically speaking, we only need $I - \boldsymbol{\Phi}_{hh} \succeq mI$, but since we want this to hold for any choice of $h$, we need the condition to apply to the entire $\boldsymbol{\Phi}$ matrix.

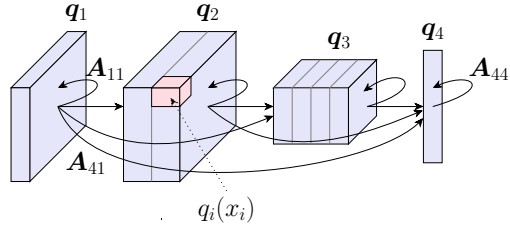

Figure 2: Illustration of a possible deep convolutional Boltzmann machine, where the monotonicity structure can still be enforced.

*Proof.* As the monotonicity condition of the monotone DEQ is assumed in the proposition, the proof of the proposition rests entirely in showing that the softmax operator is given by $\text{prox}_f^1$ for some CCP $f$. Specifically, as shown in Krähenbühl & Koltun (2013), this is the case for

$$f(z) = \sum_i z_i \log z_i - \frac{1}{2}\|z\|_2^2 + \mathbb{I}\left\{\sum_i z_i = 1, \; z_i \geq 0\right\} \tag{8}$$

i.e., the restriction of the entropy minus squared norm to the simplex (note that even though we are *subtracting* a squared norm term it is straightforward to show that this function is convex since the second derivatives are given by $1/z_i - 1$, which is always non-negative over its domain). □

### 3.3 Practical considerations when modeling mDBMs

The construction in Section 3.1 guarantees monotonicity of the resulting pairwise probabilistic model. However, instantiating the model in practice, where the variables represent hidden units of a deep architecture (i.e., representing multi-channel image tensors with pairwise potentials defined by convolutional operators), requires substantial subtlety and care in implementation. In this setting, we do not want to actually represent $\boldsymbol{A}$ explicitly, but rather determine a method for *multiplying* $\boldsymbol{A}\boldsymbol{v}$ and $\boldsymbol{A}^T\boldsymbol{v}$ for some vector $\boldsymbol{v}$ (as we see in Section 3.2, this is all that is required for the parallel mean-field inference method we propose). This means that certain blocks of $\boldsymbol{A}$ are typically parameterized as convolutional layers, with convolution and transposed convolution operators as the main units of computation.

More specifically, we typically want to *partition* the full set of hidden units into some $K$ distinct sets

$$\boldsymbol{q} = \begin{bmatrix} \boldsymbol{q}_1 & \boldsymbol{q}_2 & \cdots & \boldsymbol{q}_K \end{bmatrix}^T \tag{9}$$

where e.g., $\boldsymbol{q}_i$ would be best represented as a height $\times$ width $\times$ groups $\times$ cardinality tensor (i.e., a collection of multiple hidden units corresponding to different locations in a typical deep network hidden layer). Note that here $\boldsymbol{q}_i$ is *not* the same as $q_i(x_i)$, but rather the collection of *many* different individual variables. These $\boldsymbol{q}_i$ terms can be related to each other via different operators, and a natural manner of parameterizing $\boldsymbol{A}$, in this case, is as an interconnected set of convolutional or dense operators. To represent the pairwise interactions, we can create a similarly-factored matrix $\boldsymbol{A}$, e.g., one of the form

$$\boldsymbol{A} = \begin{bmatrix} \boldsymbol{A}_{11} & 0 & \cdots & 0 \\ \boldsymbol{A}_{21} & \boldsymbol{A}_{22} & \cdots & 0 \\ \vdots & \vdots & \ddots & \vdots \\ \boldsymbol{A}_{K1} & \boldsymbol{A}_{K2} & \cdots & \boldsymbol{A}_{KK} \end{bmatrix} \tag{10}$$

where e.g., $\boldsymbol{A}_{ij}$ is a (possibly strided) convolution mapping between the tensors representing $\boldsymbol{q}_j$ and $\boldsymbol{q}_i$. In this case, we emphasize that $\boldsymbol{A}_{ij}$ is not the kernel matrix that one "slides" along the variables. Instead, $\boldsymbol{A}_{ij}$ is the linear mapping as if we write the convolution as a matrix-matrix multiplication. For example, a 2D convolution with stride 1 can be expressed as a doubly block circulant matrix (the case is more complicated when different striding is allowed). This parametrization is effectively a *general* Boltzmann machine, since each random variable in Equation (9) can interact with any other variables except for itself. Varying $\boldsymbol{A}_{ij}$,

the formulation in Equation (10) is rich enough for any type of architecture including convolutions, fully-connected layers, and skip-connections, etc.

An illustration of one possible network structure is shown in Figure 2. To give a preliminary introduction to our implementation, let us denote the convolution filter as $\boldsymbol{F}$, and its corresponding matrix form as $\boldsymbol{A}$. While it is possibly simpler to directly compute $\boldsymbol{A}^\top \boldsymbol{A} \boldsymbol{q}$, $\boldsymbol{A}$ usually has very high dimensions even if $\boldsymbol{F}$ is small. Instead, our implementation computes $\text{CONVTRANSPOSE}(\boldsymbol{F}, \text{CONV}(\boldsymbol{F}, \boldsymbol{q}))$, modulo using the correct striding and padding. The block diagonal element of $\boldsymbol{A}^\top \boldsymbol{A}$ has smaller dimension and can be computed directly as a $1 \times 1$ convolution. The precise details of how one computes the block diagonal elements of $\boldsymbol{A}^T \boldsymbol{A}$, and how one normalizes the proper diagonal blocks (which, we emphasize, still just requires computing the singular values of matrices whose size is the cardinality of a single $q_i(x_i)$) are somewhat involved, so we defer a complete description to the Appendix (and accompanying code). The larger takeaway message, though, is that *it is possible to parameterize complex convolutional multi-scale Boltzmann machines, all while ensuring monotonicity.*

### 3.4   Efficient parallel solving for the mean-field fixed point

Although the monotonicity of $\boldsymbol{\Phi}$ guarantees the existence of a unique solution, it does not necessarily guarantee that the simple iteration

$$\boldsymbol{q}_h^{(t)} = \text{softmax}(\boldsymbol{\Phi}_{hh}\boldsymbol{q}_h^{(t-1)} + \boldsymbol{\Phi}_{ho}\boldsymbol{x}_o + \boldsymbol{b}_h) \tag{11}$$

will converge to this solution. Instead, to guarantee convergence, one needs to apply the *damped* iteration (see, e.g. (Winston & Kolter, 2020))

$$\boldsymbol{q}_h^{(t)} = \text{prox}_f^\alpha \left( (1-\alpha)\boldsymbol{q}_h^{(t-1)} + \alpha(\boldsymbol{\Phi}_{hh}\boldsymbol{q}_h^{(t-1)} + \boldsymbol{\Phi}_{ho}\boldsymbol{x}_o + \boldsymbol{b}_h) \right). \tag{12}$$

The damped forward-backward iteration converges linearly to the unique fixed point if $\alpha \leq 2m/L^2$, assuming $I - \boldsymbol{\Phi}$ is $m$-strongly monotone and $L$-Lipschitz (Ryu & Boyd, 2016). Crucially, this update can be formed *in parallel* over all the variables in the network: we do not require a coordinate descent approach as is typically needed by mean-field inference.

The key issue, though is that while $\text{prox}_f^1(x) = \text{softmax}(x)$ for $f$ defined as in Equation (8), in general, this does not hold for $\alpha \neq 1$. Indeed, for $\alpha \neq 1$, there is no closed-form solution to the proximal operation, and computing the solution is substantially more involved. Specifically, computing this proximal operator involves solving the optimization problem

$$\text{prox}_f^\alpha(x) = \arg\min_z \ \frac{1}{2}\|x - z\|_2^2 + \alpha \sum_i z_i \log z_i - \frac{\alpha}{2}\|z\|_2^2 \quad \text{s.t} \ \sum_i z_i = 1, \ z \geq 0. \tag{13}$$

The following theorem, proved in the Appendix, characterizes the solution to this problem for $\alpha \in (0, 1)$ (although it is also possible to compute solutions for $\alpha > 1$, this is not needed in practice, as it corresponds to a "negatively damped" update, and it is typically better to simply use the softmax update in such cases).

**Theorem 3.2.** *Given $f$ as defined in Equation (8), $\alpha \in (0, 1)$, and $x \in \mathbb{R}^k$, the proximal operator $\text{prox}_f^\alpha(x)$ is given by*

$$\text{prox}_f^\alpha(x)_i = \frac{\alpha}{1-\alpha} W \left( \frac{1-\alpha}{\alpha} \exp \left( \frac{x_i - \alpha + \lambda}{\alpha} \right) \right),$$

*where $\lambda \in \mathbb{R}$ is the unique solution chosen to ensure that the resulting $\sum_i \text{prox}_f^\alpha(x_i) = 1$, and where $W(\cdot)$ is the principal branch of the Lambert $W$ function.*

In practice, however, this is not the most numerically stable method for computing the proximal operator, especially for small $\alpha$, owing to the large term inside the exponential. Computing the proximal operation efficiently is somewhat involved, though briefly, we define the alternative function

$$g(y) = \log \frac{\alpha}{1-\alpha} W \left( \frac{1-\alpha}{\alpha} \exp \left( \frac{y}{\alpha} - 1 \right) \right) \tag{14}$$

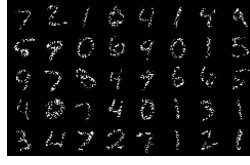
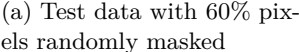

(a) Test data with 60% pixels randomly masked

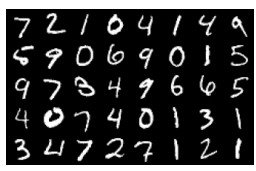

(b) Original image

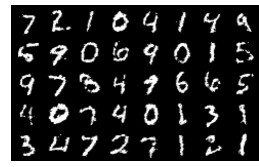

(c) Imputation with 60% pixels randomly masked using mDBM

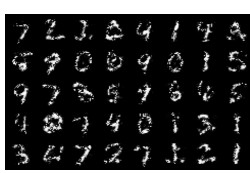

(d) Imputation with 60% pixels randomly masked using RBM

Figure 3: MNIST pixel imputation using mDBM (bottom left) and deep RBM (bottom right), where the RBM test results are generated using mean-field inference instead of Gibbs sampling.

and show how to directly compute $g(y)$ using Halley's method (note that Halley's method is also the preferred manner to computing the Lambert W function itself numerically (Corless et al., 1996)). It updates $x_{n+1} = x_n - \frac{2f(x_n)f'(x_n)}{2(f'(x_n))^2 - f(x_n)f''(x_n)}$, and enjoys cubic convergence when the initial guess is close enough to the root. Finding the prox operator then requires that we find $\lambda$ such that $\sum_{i=1}^{k} \exp(g(x_i + \lambda)) = 1$. This can be done via (one-dimensional) root finding with Newton's method, which is guaranteed to always find a solution here, owing to the fact that this function is convex monotonic for $\lambda \in (-\infty, 1)$. We can further compute the gradients of the $g$ function and of the proximal operator itself via implicit differentiation (i.e., we can do it analytically without requiring unrolling the Newton or Halley iteration). We describe the details in the appendix and include an efficient PyTorch function implementation in the supplementary material.

**Comparison to Winston & Kolter (2020)**  Although this work uses the same monotonicity constraint as in Winston & Kolter (2020), our result further requires the linear module $\mathbf{\Phi}$ to be hollow, and extend their work to the softmax nonlinear operator as well. These extensions introduce significant complications, but also enable us to interpret our network as a probabilistic model, while the network in Winston & Kolter (2020) cannot.

### 3.5 Training considerations

Finally, we discuss approaches for training mDBMs, exploiting their efficient approach to mean-field inference. Probabilistic models are typically trained via approximate likelihood maximization, and since the mean-field approximation is based upon a particular likelihood approximation, it may seem most natural to use this same approximation to train parameters. In practice, however, this is often a suboptimal approach. Specifically, because our forward inference procedure ultimately uses mean-field inference, it is better to train the model directly to output the correct marginals, *when running this mean-field procedure.* This is known as a marginal-based loss (Domke, 2013). In the context of mDBMs, this procedure has a particularly convenient form, as it corresponds roughly to the "typical" training of DEQ.

In more detail, suppose we are given a sample $\mathbf{x} \in \mathcal{X}$ (i.e., at training time the entire sample is given), along with a specification of the "observed" and "hidden" sets, $o$ and $h$ respectively. Note that the choice of observed and hidden sets is potentially up to the algorithm designer, and can effectively allow one to train our model in a "self-supervised" fashion, where the goal is to predict some unobserved components from others. In practice, however, one typically wants to design hidden and observed portions congruent with the eventual use of the model: e.g., if one is using the model for classification, then at training time it makes sense for the label to be "hidden" and the input to be "observed."

Given this sample, we first solve the mean-field inference problem to find $\boldsymbol{q}_h^{\star}(\mathbf{x}_h)$ such that

$$\boldsymbol{q}_h^{\star} = \text{softmax}\left(\mathbf{\Phi}_{hh}\boldsymbol{q}_h^{\star} + \mathbf{\Phi}_{ho}\mathbf{x}_o + \boldsymbol{b}_h\right). \tag{15}$$

For this sample, we know that the true value of the hidden states is given by $\mathbf{x}_h$. Thus, we can apply some loss function $\ell(\boldsymbol{q}_h^{\star}, \mathbf{x}_h)$ between the prediction and true value, and update parameters of the model

$\theta = \{\boldsymbol{A}, \boldsymbol{b}\}$ using their gradients

$$\frac{\partial \ell(\boldsymbol{q}_h^\star, \mathbf{x}_h)}{\partial \theta} = \frac{\partial \ell(\boldsymbol{q}_h^\star, \mathbf{x}_h)}{\partial \boldsymbol{q}_h^\star} \frac{\partial \boldsymbol{q}_h^\star}{\partial \theta} = \frac{\partial \ell(\boldsymbol{q}_h^\star, \mathbf{x}_h)}{\partial \boldsymbol{q}_h^\star} \left( I - \frac{\partial g(\boldsymbol{q}_h^\star)}{\partial \boldsymbol{q}_h^\star} \right)^{-1} \frac{\partial g(\boldsymbol{q}_h^\star)}{\partial \theta} \tag{16}$$

with

$$g(\boldsymbol{q}_h^\star) \equiv \mathrm{prox}_f^\alpha \left( (1-\alpha)\boldsymbol{q}_h^* + \alpha(\boldsymbol{\Phi}_{hh}\boldsymbol{q}_h^* + \boldsymbol{\Phi}_{ho}\boldsymbol{x}_o + \boldsymbol{b}_h) \right)$$

and where the last equality comes from the standard application of the implicit function theorem as typical in DEQs or monotone DEQs. The key to gradient computation is noticing that in Equation (16), we can rearrange:

$$\boldsymbol{u} \triangleq \frac{\partial \ell(\boldsymbol{q}_h^\star, \mathbf{x}_h)}{\partial \boldsymbol{q}_h^\star} \left( I - \frac{\partial g(\boldsymbol{q}_h^\star)}{\partial \boldsymbol{q}_h^\star} \right)^{-1} \implies \boldsymbol{u} = \boldsymbol{u}\frac{\partial g(\boldsymbol{q}_h^\star)}{\partial \boldsymbol{q}_h^\star} + \frac{\partial \ell(\boldsymbol{q}_h^*, \boldsymbol{x}_h)}{\partial \boldsymbol{q}_h^*},$$

which is just another fixed-point problem. Here all the partial derivatives can be handled by auto-differentiation (with the correct backward hook for $\mathrm{prox}_f^\alpha$), and the details exactly mirror that of Winston & Kolter (2020), also see Algorithm 2.

As a final note, we also mention that owing to the restricted range of weights allowed by the monotonicty constraint, the actual output marginals $q_i(x_i)$ are often more uniform in distribution than desired. Thus, we typically apply the loss to a scaled marginal

$$\tilde{q}_i(x_i) \propto q_i(x_i)^{\tau_i} \tag{17}$$

where $\tau_i \in \mathbb{R}_+$ is a variable-dependent learnable temperature parameter. Importantly, we emphasize that this is *only* done after convergence to the mean-field solution, and thus only applies to the marginals to which we apply a loss: the actual internal iterations of mean-field cannot have such a scaling, as it would violate the monotonicity condition.

---

**Algorithm 1** FORWARDITERATION

---

**Require:** Observed RV $\boldsymbol{x}_o$, parameters $\boldsymbol{\Phi}, \boldsymbol{b}$, damp parameter $\alpha \in (0,1)$.
  Find the fixed point $\boldsymbol{q}_h^*$ to

$$\boldsymbol{q}_h^{(t)} = \mathrm{prox}_f^\alpha \left( (1-\alpha)\boldsymbol{q}_h^{(t-1)} + \alpha(\boldsymbol{\Phi}_{hh}\boldsymbol{q}_h^{(t-1)} + \boldsymbol{\Phi}_{ho}\boldsymbol{x}_o + \boldsymbol{b}_h) \right)$$

---

**Algorithm 2** BACKWARDITERATION

---

**Require:** Loss function $\ell$, fixed point $\boldsymbol{q}_h^*$, true value $\boldsymbol{x}_h$ parameters $\theta$.
  Find the fixed point $\boldsymbol{u}^*$ to
$$\boldsymbol{u} = \boldsymbol{u}\frac{\partial g(\boldsymbol{q}_h^*)}{\partial \boldsymbol{q}_h^*} + \frac{\partial \ell(\boldsymbol{q}_h^*, \boldsymbol{x}_h)}{\partial \boldsymbol{q}_h^*}$$

Compute the final Jacobian-vector product as

$$\frac{\partial \ell(\boldsymbol{q}_h^\star, \mathbf{x}_h)}{\partial \theta} = \boldsymbol{u}^* \frac{\partial g(\boldsymbol{q}_h^*)}{\partial \theta}$$

---

A simple overview of the algorithm is demonstrated in Algorithm 3. The task is to jointly predict the image label and fill the top-half of the image given the bottom-half. For inference, the process is almost the same as training, except we don't update the parameters.

---

**Algorithm 3** TRAINING

---

**Require:** Damp parameter $\alpha \in (0, 1)$, neural network parameters $\mathbf{\Phi}, \boldsymbol{b}$, weight on classification loss $w$.
   **for** each epoch **do**
      **for** $(\boldsymbol{p}, y)$ in data **do**
         Let $\boldsymbol{x}_h = \{\boldsymbol{p}_h, y\}$ where $\boldsymbol{p}_h$ is the top-half of the image $\boldsymbol{p}$ and $y$ is the label. Let $\boldsymbol{x}_o$ be the bottom-half
         of the image $\boldsymbol{p}_o$.
         $\boldsymbol{q}_h^* = \text{FORWARDITERATION}(\boldsymbol{x}_o, \mathbf{\Phi}, \boldsymbol{b}, \alpha)$, notice that $\boldsymbol{x}_h$ is not revealed to the network here.
         Get the estimated top half of the image and label $\{\hat{\boldsymbol{p}}_h, \hat{y}\} = \boldsymbol{q}_h^*$.
         Calculate imputation loss and classification loss $(1 - w)\ell_r(\hat{\boldsymbol{p}}_h, \boldsymbol{p}_h) + w\ell_c(\hat{y}, y)$.
         Update $\mathbf{\Phi}, \boldsymbol{b}$ using Algorithm 2.
      **end for**
   **end for**

---

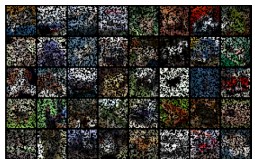 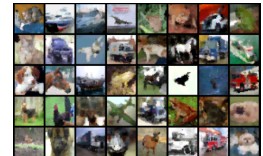 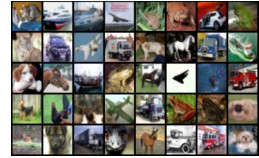 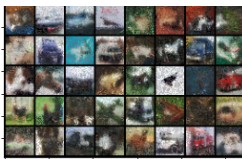

(a) Test data has 50% pixels randomly masked    (b) Imputed pixel inference (without injection labels)    (c) Original image    (d) Imputation with deep RBM.

Figure 4: CIFAR-10 pixel imputation using mDBM and deep RBM

## 4 Experimental evaluation

As a proof of concept, we evaluate our proposed mDBM on the MNIST and CIFAR-10 datasets. We demonstrate how to jointly model missing pixels and class labels conditioned on only a subset of observed pixels. On MNIST, we compare mDBM to mean-field inference in a traditional deep RBM. Despite being small-scale tasks, the goal here is to demonstrate joint inference and learning over what is still a reasonably-sized joint model, considering the number of hidden units. Nonetheless, the current experiments are admittedly largely a *demonstration* of the proposed method rather than a full accounting of its performance.

We also show how our mean-field inference method compares to those proposed in prior works. On the joint imputation and classification task, we train models using our updates and the updates proposed by Krähenbühl & Koltun (2013) and Baqué et al. (2016), and perform mean-field inference in each model using all three update methods, with and without the monotonicity constraint.

**mDBM and deep RBM on MNIST** For the joint imputation and classification task, we randomly mask each pixel independently with probability 60%, such that in expectation only 40% of the pixels are observed. The original MNIST dataset has one channel representing the gray-scale intensity, ranging between 0 and 1. We adopt the strategy of Van Oord et al. (2016) to convert this continuous distribution to a discrete one. We bin the intensity evenly into 4 categories $\{0, \ldots, 3\}$, and for each channel use a one-hot encoding of the category so that the input data has shape $4 \times 28 \times 28$. We remark that the number of categories is chosen arbitrarily and can be any integer. Additional details are given in the appendix.

The mDBM and deep RBM trained on the joint imputation and classification task obtain test classification accuracy of 92.95% and 64.23%, respectively. Pixel imputation is shown in Figure 3. We see that the deep RBM is not able to impute the missing pixels well, while the mDBM can. Importantly, we note however that for an apples-to-apples comparison, the test results in the RBM are generated using mean-field inference. The image imputation of RBM runs block mean-field updates of 1000 steps and the classification runs 2 steps, and increasing number of iterations does not improve test performance significantly. The RBM also admits efficient Gibbs sampling, which performs much better and is detailed in the appendix.

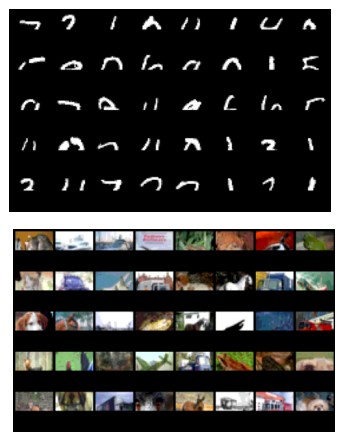 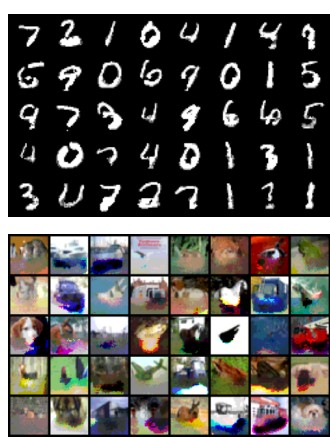 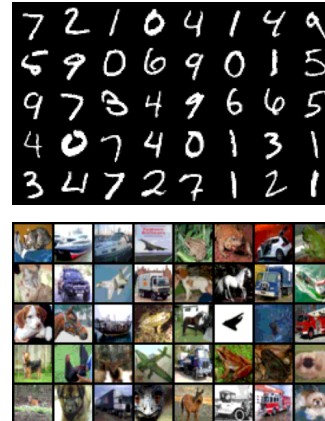

Figure 5: MNIST and CIFAR10 pixel imputation using mDBM, when only the top half is shown to the model. **Left**: observed images; **middle**: imputation results; **right**: original images.

Table 1: Squared $\ell_2$ error (standard deviation) for MNIST image imputation. Images are observed $20\%, 40\%, 60\%, 80\%$ respectively. RBM outputs are bucketized into 10 bins. The errors are averaged over the whole dataset and the number of bins. The experiments are executed 5 times with independent random masks and the same models. Standard deviations are calculated across 5 runs.

| Method \ Observation | 0.2 | 0.4 | 0.6 | 0.8 |
|---|---|---|---|---|
| mDBM | 53.310 (0.0776) | 23.330 (0.0204) | 13.140 (0.0234) | 5.936 (0.0102) |
| RBM | 53.086 (0.0556) | 36.596 (0.0417) | 22.746 (0.0234) | 10.564 (0.0186) |

We report the image imputation $\ell_2$ loss on MNIST in Table 1. We randomly mask $p = \{0.2, 0.4, 0.6, 0.8\}$ portion of the inputs. For each $p$, the experiments are conducted 5 times where each run independently chooses the random mask. The model is trained to impute images given 40% pixels and is fixed throughout the experiments. Since the RBMs model Bernoulli random variables whereas mDBMs model distributions over a set of one-hot variables, we "bucketize"[3] the outputs of RBMs into bins of size 10. The $\ell_2$ norm of the difference between bucketized imputations and the original images is computed. The norms are then divided by the number of bins and averaged over the whole MNIST dataset. In this way, we get $\{\mu_1, \ldots, \mu_5\}$ where each $\mu_i$ is the average $\ell_2$ reconstruction error over the whole dataset, and the standard deviations are calculated over $\{\mu_1, \ldots, \mu_5\}$. Our proposed method has a clear advantage over RBMs.

We additionally evaluate mDBM on a task in which random $14 \times 14$ patches are masked. To obtain good performance on this task requires lifting the monotonicity constraint; we find that mDBM converges regardless (see appendix). mDBM can also extrapolate reasonably well, see Figure 5.

**mDBM and deep RBM on CIFAR-10**  We evaluate mDBM on an analogous task of image pixel imputation and label prediction on CIFAR-10. Model architecture and training details are given in the appendix. With 50% of the pixels observed, the model obtains 58% test accuracy and can impute the missing pixels effectively (see Figure 4). The baseline deep RBM is trained to impute the missing pixels using CD-1 with number of neurons 3072-500-100. The imputation error is reported in Table 2 and the experiments are conducted in the same fashion as those on MNIST. Contrary to grayscale MNIST, RBM outputs are bucketized into 10 bins for each of the RGB channels on CIFAR-10. mDBMs also take bucketized images as inputs where each of the RGB channels is bucketized into 10 bins.

---

[3]That is, if the number of bins is 2 and the RBM outputs a probability $p$, the bucketized output is 0 if $p < 0.5$ and 1 otherwise.

Table 2: Squared $\ell_2$ error (standard deviation) for CIFAR10 image imputation. Images are observed $20\%, 40\%, 60\%, 80\%$ respectively. RBM outputs are bucketized into 10 bins for each of the RGB channels. mDBMs also take bucketized images as inputs where each of the RGB channels is bucketized into 10 bins. The errors are averaged over the whole dataset and the number of bins. The experiments are executed 5 times with independent random masks and the same models. Standard deviations are calculated across 5 runs.

| Observation Method | 0.2 | 0.4 | 0.6 | 0.8 |
|---|---|---|---|---|
| mDBM | 77.439 (0.0281) | 48.496 (0.0166) | 29.749 (0.0241) | 14.077 (0.0143) |
| RBM | 139.375 (0.0231) | 103.615 (0.0315) | 68.845 (0.0219) | 34.339 (0.0291) |

Table 3: Relative update residual when monotonicity is enforced.

| Inference Train | Krähenbühl | Baqué | mDBM |
|---|---|---|---|
| Krähenbühl | 0.0004 | 0.0061 | 0.0024 |
| Baqué | 1.250 | 0.0059 | 0.0024 |
| mDBM | 1.144 | 0.0057 | 0.0017 |

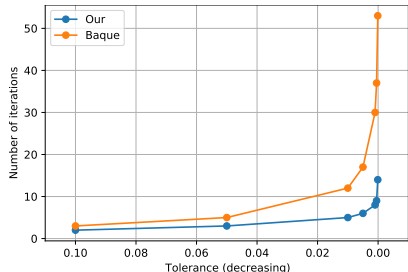

Figure 6: Convergence speed of inference methods on a model trained with Krähenbühl's updates.

**Comparison of inference methods** We conduct several experiments comparing our mean-field inference method to those proposed by Krähenbühl & Koltun (2013) and Baqué et al. (2016), denoted as Krähenbühl's and Baqué's respectively. While a full description of these methods and the experiments is left to the appendix, we highlight some of the key findings here. We train models using the three different update methods: ours, Krähenbühl's and Baqué's; we then perform inference using all three methods as well. A comparison to the regularized Frank-Wolfe method raised by Lê-Huu & Alahari (2021) can be found in the appendix.

Table 3 shows the relative update residuals $\|\boldsymbol{q}_h^{(t+1)} - \boldsymbol{q}_h^{(t)}\|/\|\boldsymbol{q}_h^{(t)}\|$ after 100 steps of each inference method on each model. We observe that Krähenbühl's method diverges when the model was not trained using the corresponding updates, whereas Baqué's and ours converge on all three models, with our method converging more quickly. The improved convergence speed can also be seen in Figure 6). However, note that Baqué's method is not guaranteed to converge to the true mean-field fixed-point. As we show in the appendix (Figure 11c), on an untrained model our method converges to the true fixed-point while Baqué's does not.

**Future directions** It is extremely useful to consider fundamentally different restrictions as have been applied to meanfield inference and graphical models in the past, and our work can lead to a number of interesting directions: (1) Theorem 3.1 is only a sufficient but not necessary condition for monotonicity. Improving this could potentially make our current monotone model much more expressive. (2) In Theorem 3.1, the parameter $m > 0$ describes how monotone the model is. Is it possible to use a $m < 0$ to ensure that the model is "boundedly non-monotone", but still enjoys favorable convergence property? (3) Our model currently only learns conditional probability. Is it possible to make it model joint probability efficiently? One way is to mimic PixelCNN: let $P(x_1^n) = \prod_{i=1}^n P(x_n|x_1^{n-1})$. This is inefficient for us in both inference and training, is there a way to improve? (4) Although we have a fairly efficient implementation of $\text{prox}_f^\alpha$, it is still slower than normal nonlinearities like ReLU or softmax. Is there a way to efficiently scale mDBMs? (5) Tsuchida & Ong (2022) explores the connection between PCA and DEQs, what are other probabilistic models that can also be expressed within the DEQ framework? (6) Bechmark mDBM image imputations together with Yoon et al. (2018); Li et al. (2019); Mattei & Frellsen (2019); Richardson et al. (2020).

## 5 Conclusion

In this work, we give a monotone parameterization for general Boltzmann machines and connect its mean-field fixed point to a monotone DEQ model. We provide a mean-field update method that is proven to be globally convergent. Our parameterization allows for full parallelization of mean-field updates without restricting the potential function to be concave, thus addressing issues with prior approaches. Moreover, we allow complicated and hierarchical structures among the variables and show how to efficiently implement them. For parameter learning, we directly optimize the marginal-based loss over the mean-field variational family, circumventing the intractability of computing the partition function. Our model is evaluated on the MNIST and CIFAR-10 datasets for simultaneously predicting with missing data and imputing the missing data itself. As a demonstration of concept, we also deliver several illustrations of interesting future directions.

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

## A Appendix

### A.1 Deferred proofs

**Theorem 3.2.** *Given $f$ as defined in Equation (8), $\alpha \in (0,1)$, and $x \in \mathbb{R}^k$, the proximal operator $\text{prox}_f^\alpha(x)$ is given by*

$$\text{prox}_f^\alpha(x)_i = \frac{\alpha}{1-\alpha} W\left(\frac{1-\alpha}{\alpha}\exp\left(\frac{x_i - \alpha + \lambda}{\alpha}\right)\right),$$

*where $\lambda \in \mathbb{R}$ is the unique solution chosen to ensure that the resulting $\sum_i \text{prox}_f^\alpha(x_i) = 1$, and where $W(\cdot)$ is the principal branch of the Lambert W function.*

*Proof.* By definition, the proximal operator induced by $f$ (the same $f$ in Equation (8)) and $\alpha$ solves the following optimization problem:

$$\min_z \quad \frac{1}{2}\|x - z\|^2 + \alpha\sum_i z_i \log z_i - \frac{\alpha}{2}\|z\|^2$$

$$\text{s.t.} \quad z_i \geq 0, \ i = 1, \ldots, d,$$

$$\sum_i z_i = 1$$

of which the KKT condition is

$$-x_i + z_i + \alpha + \alpha\log z_i - \alpha z_i + \lambda - \mu_i = 0, \text{for } i \in [d]$$

$$\mu_i \geq 0, \quad z_i \geq 0, \quad \sum_{i\in[d]}\mu_i z_i = 0, \quad \sum_{i=1}^d z_i = 1.$$

We have that $\mu_i = 0$ is feasible and the first equation of the above KKT condition can be massaged as

$$-x_i + z_i + \alpha + \alpha\log z_i - \alpha z_i + \lambda - \mu_i = 0$$

$$\iff (1-\alpha)z_i + \alpha\log z_i = x_i - \alpha - \lambda$$

$$\iff \frac{(1-\alpha)z_i + \alpha\log z_i}{\alpha} = \frac{x_i - \alpha - \lambda}{\alpha}$$

$$\iff \exp\left(\frac{(1-\alpha)z_i + \alpha\log z_i}{\alpha}\right) = \exp\left(\frac{x_i - \alpha - \lambda}{\alpha}\right)$$

$$\iff z_i \exp\left(\frac{1-\alpha}{\alpha}z_i\right) = \exp\left(\frac{x_i - \alpha - \lambda}{\alpha}\right)$$

$$\iff \frac{1-\alpha}{\alpha}z_i \exp\left(\frac{1-\alpha}{\alpha}z_i\right) = \frac{1-\alpha}{\alpha}\exp\left(\frac{x_i - \alpha - \lambda}{\alpha}\right)$$

$$\iff \frac{1-\alpha}{\alpha}z_i = W\left(\frac{1-\alpha}{\alpha}\exp\left(\frac{x_i - \alpha - \lambda}{\alpha}\right)\right)$$

where $W$ is the lambert W function. Notice here $z_i > 0$. Our primal problem is convex and Slater's condition holds. Hence, we conclude that

$$z_i = \frac{\alpha}{1-\alpha}W\left(\frac{1-\alpha}{\alpha}\exp\left(\frac{x_i - \alpha - \lambda}{\alpha}\right)\right).$$

$\square$

### A.2 Convolution network

It is clear that the monotone parameterization in Section 3 directly applies to fully-connected networks, and all the related quantities can be calculated easily. Nonetheless, the real power of the DEQ model comes in

when we use more sophisticated linear operators like convolutions. In the context of Boltzmann machines, the convolution operator gives edge potentials beneficial structures. For example, when modeling the joint probability of pixels in an image, it is intuitive that only the nearby pixels depend closely on each other.

Let $A \in \mathbb{R}^{k \times k \times r \times r}$ denote a convolutional tensor with kernel size $r$ and channel size $k$, let $x$ denote some input. For a convolution with stride 1, the block diagonal elements of $A^T A$ simply form a $1 \times 1$ convolution. In particular, we apply the convolutions

$$- A^T(A(x)) + \tilde{A}(x) \tag{18}$$

where $\tilde{A}$ is a $1 \times 1$ convolution given by

$$\tilde{A}[:,:] = \sum_{i,j} A[:,:,i,j]^T A[:,:,i,j]. \tag{19}$$

We can normalize by the spectral norm of $\tilde{A}$ term to ensure strong monotonicity. Since $\tilde{A}$ can be rewritten as a $k \times k$ matrix and $k$ is usually small, its spectral norm can be easily calculated.

It takes more effort to work out convolutions with stride other than 1. Specifically, the block diagonal terms do not form a $1 \times 1$ convolution anymore, instead, the computation varies depending on the location. It is easier to see the computation directly in the accompanying code.

**Grouped channels** It is crucial to introduce the concept of grouped channels, which allows us to represent multiple categorical variables in a single location, such as the three categorical variables representing the three (binned) color channels of an RGB pixel. In this case, each of the three RGB channels will be represented by a different group of $k$ channels representing the $k$ bins. The grouping is achieved by having the nonlinearity function (softmax) applied to each group separately. We remark that the convolutions themselves are *not* grouped, otherwise none of the red pixels would interact with green or blue pixels, etc. Instead, we want all RGB channels to interact with each other (except that channel $i$ at position $(j, k)$ does not interact with itself). That means in Equation (3), the blkdiag($\hat{A}^T \hat{A}$) is grouped in the following way. Recall that this block diagonal term has element of size $k_i \times k_i$ for $i \in [n]$. This parameterization has only 1 group. With $g$ groups, the element of the block diagonal matrix then has size $k_{i_1} \times k_{i_1}, \ldots, k_{i_g} \times k_{i_g}$ for $i \in [n]$, where $\sum_{j \in [g]} k_{i_j} = k_i$. We also observe empirically that grouping the latent variables improves the performance.

### A.3 Efficient computation of $\operatorname{prox}_f^\alpha$

The solution to the proximal operator in damped forward iteration given in Theorem 3.2 involves the Lambert W function, which does not attain an analytical solution. In this section, we show how to efficiently calculate the nonlinearity $\sigma(x_i)$, as well as its Jacobian matrix for backward iteration.

Let $f(y) = \frac{\alpha}{1-\alpha} W\left(\frac{1-\alpha}{\alpha} \exp\left(\frac{y}{\alpha} - 1\right)\right)$, and we have

$$x \quad = \quad \log f(y) \quad = \quad \log \frac{\alpha}{1-\alpha} \quad + \quad \log e^{y/\alpha - 1} \quad + \quad \log \frac{1-\alpha}{\alpha} \quad - \quad W\left(\frac{1-\alpha}{\alpha} \exp\left(\frac{y}{\alpha} - 1\right)\right),$$

where the last equality uses the identity $\log(W(x)) = \log x - W(x)$. Rewrite $W\left(\frac{1-\alpha}{\alpha} \exp\left(\frac{y}{\alpha} - 1\right)\right) = f(y)\frac{1-\alpha}{\alpha}$ and massage the terms, we have that solving $\log f(y)$ is equivalent to finding the root of

$$h(x) = y - \alpha - e^x(1 - \alpha) - \alpha x.$$

Direct calculation shows that $h'(x) = -\alpha - (1-\alpha)e^x$ and $h''(x) = -(1-\alpha)e^x$. Note here $y$ is the input and it is known to us, and $x$ is a scalar. Hence we can efficiently solve the root finding problem using Halley's method. For backpropagation, we need $\frac{dx}{dy}$, which can be computed by implicit differentiation:

$$h(x) = y - \alpha - e^x(1 - \alpha) - \alpha x = 0$$

$$\implies \frac{dx}{dy} = \frac{1}{\alpha + (1-\alpha)e^x} = \frac{1}{y - \alpha x}.$$

Now we can find $\lambda$ s.t $\sum_i z_i = 1$ using Newton's method on $g(\lambda) = \sum_i e^{\log(f(x_i + \lambda))} - 1 = 0$. Note this is still a one-dimensional optimization problem. A direct calculation shows that $\frac{dg}{d\lambda} = \sum_i e^{\log(f(x_i+\lambda))} \frac{d\log(f(x_i+\lambda))}{d\lambda}$, and above we have already calculated that

$$\frac{d\log(f(x_i + \lambda))}{d\lambda} = \frac{dx^*}{dy} = \frac{1}{y + \lambda - \alpha x}.$$

For backward computation, by the chain rule, we have:

$$\frac{de^{\log f(x_i+\lambda)}}{dx_i} = e^{\log f(x_i+\lambda)} \frac{d\log(f(x_i + \lambda))}{dx_i}$$

$$= e^{\log f(x_i+\lambda)} \frac{1 + d\lambda/dx_i}{x_i + \lambda - \alpha \log(f(x_i + \lambda))},$$

where the last step is derived by implicit differentiation. Now to get $d\lambda/dx_i$, notice that by applying the implicit function theorem on $p(x, \lambda(x)) = \sum_i e^{\log(f(x_i+\lambda))} - 1 = 0$, we get

$$\frac{d\lambda}{dx_i} = -\left(\frac{dp}{d\lambda}\right)^{-1} \frac{dp}{dx_i}.$$

Thus we have all the terms computed, which finishes the derivation.

# B    Additional Experiments and Details

Here we provide the model architectures and experiment details omitted in the main text.

## B.1    Details

**Model architectures**    For MNIST experiments (except for the extrapolation), using the notation in Equation (10), the mDBM consists of a 4-layer deep monotone DEQ with the following structure:

$$\begin{bmatrix} \boldsymbol{A}_{11} & 0 & 0 & 0 \\ \boldsymbol{A}_{21} & \boldsymbol{A}_{22} & 0 & 0 \\ \boldsymbol{A}_{31} & \boldsymbol{A}_{32} & \boldsymbol{A}_{33} & 0 \\ 0 & 0 & \boldsymbol{A}_{43} & \boldsymbol{A}_{44} \end{bmatrix},$$

where $\boldsymbol{A}_{11}$ is a $20 \times 20 \times 3 \times 3$ convolution, $\boldsymbol{A}_{22}$ is a $40 \times 40 \times 3 \times 3$ convolution, $\boldsymbol{A}_{21}$ is a $40 \times 20 \times 3 \times 3$ convolution with stride 2, $\boldsymbol{A}_{33}$ is a $80 \times 80 \times 3 \times 3$ convolution, $\boldsymbol{A}_{31}$ is a $80 \times 20 \times 3 \times 3$ convolution with stride 4, $\boldsymbol{A}_{32}$ is a $80 \times 40 \times 3 \times 3$ convolution with stride 2, $\boldsymbol{A}_{43}$ is a $(80 \cdot 7 \cdot 7) \times 10$ dense linear layer, and $\boldsymbol{A}_{44}$ is a $10 \times 10$ dense linear layer. The corresponding variable $\boldsymbol{q}$ as in Equation (9) then has 4 elements of shape $(20 \times 28 \times 28), (40 \times 14 \times 14), (80 \times 7 \times 7), (10 \times 1)$. When applying the proximal operator to $\boldsymbol{q}$, we use $1, 10, 20, 1$ as their number of groups, respectively.

The deep RBM consists of 3-layers where the first hidden layer has 300 neurons, and the last hidden layer (representing the digits) has 10 neurons, amounting to in total 239,294 parameters. For comparison, the mDBM has 192,650 parameters.

The mDBM used on CIFAR-10 is the same as for the MNIST experiments with the following exceptions: $\boldsymbol{A}_{11}$ is a $20 \times 20 \times 3 \times 3$ convolution, $\boldsymbol{A}_{22}$ is a $24 \times 24 \times 3 \times 3$ convolution, $\boldsymbol{A}_{21}$ is a $24 \times 20 \times 3 \times 3$ convolution with stride 2, $\boldsymbol{A}_{33}$ is a $48 \times 48 \times 3 \times 3$ convolution, $\boldsymbol{A}_{31}$ is a $48 \times 20 \times 3 \times 3$ convolution with stride 4, $\boldsymbol{A}_{32}$ is a $48 \times 24 \times 3 \times 3$ convolution with stride 2, $\boldsymbol{A}_{43}$ is a $(48 \cdot 8 \cdot 8) \times 10$ dense linear layer, and $\boldsymbol{A}_{44}$ is a $10 \times 10$ dense linear layer. The corresponding variable $\boldsymbol{q}$ as in Equation (9) then has 4 elements of shape $(60 \times 32 \times 32), (24 \times 16 \times 16), (48 \times 8 \times 8), (10 \times 1)$. When applying the proximal operator to $\boldsymbol{q}$, we use $1, 6, 12, 1$ as their number of groups, respectively.

For the MNIST extrapolation experiments, we use mDBM of the following structure:

$$\begin{bmatrix} \boldsymbol{A}_{11} & 0 & 0 & 0 \\ \boldsymbol{A}_{21} & \boldsymbol{A}_{22} & 0 & 0 \\ \boldsymbol{A}_{31} & \boldsymbol{A}_{32} & \boldsymbol{A}_{33} & 0 \\ \boldsymbol{A}_{41} & \boldsymbol{A}_{42} & \boldsymbol{A}_{43} & \boldsymbol{A}_{44} \end{bmatrix},$$

where $\boldsymbol{A}_{11}$ is a $4 \times 4 \times 3 \times 3$ convolution, $\boldsymbol{A}_{22}$ is a $40 \times 40 \times 3 \times 3$ convolution, $\boldsymbol{A}_{21}$ is a $40 \times 4 \times 3 \times 3$ convolution with stride 2, $\boldsymbol{A}_{33}$ is a $80 \times 80 \times 3 \times 3$ convolution, $\boldsymbol{A}_{31}$ is a $80 \times 4 \times 3 \times 3$ convolution with stride 4, $\boldsymbol{A}_{32}$ is a $80 \times 40 \times 3 \times 3$ convolution with stride 2, $\boldsymbol{A}_{41}$ is a $(4 \cdot 28 \cdot 28) \times 100$ dense linear layer, $\boldsymbol{A}_{42}$ is a $(40 \cdot 14 \cdot 14) \times 100$ dense linear layer $\boldsymbol{A}_{43}$ is a $(80 \cdot 7 \cdot 7) \times 100$ dense linear layer, and $\boldsymbol{A}_{44}$ is a $100 \times 100$ dense linear layer. The corresponding variable $\boldsymbol{q}$ as in Equation (9) then has 4 elements of shape $(4 \times 28 \times 28), (40 \times 14 \times 14), (80 \times 7 \times 7), (100 \times 1)$. When applying the proximal operator to $\boldsymbol{q}$, we use $1, 10, 20, 10$ as their number of groups, respectively.

The model used for CIFAR-10 extrapolation has the same structure, where $\boldsymbol{A}_{11}$ is a $30 \times 30 \times 3 \times 3$ convolution, $\boldsymbol{A}_{22}$ is a $60 \times 60 \times 3 \times 3$ convolution, $\boldsymbol{A}_{21}$ is a $60 \times 30 \times 3 \times 3$ convolution with stride 2, $\boldsymbol{A}_{33}$ is a $80 \times 80 \times 3 \times 3$ convolution, $\boldsymbol{A}_{31}$ is a $80 \times 30 \times 3 \times 3$ convolution with stride 4, $\boldsymbol{A}_{32}$ is a $80 \times 60 \times 3 \times 3$ convolution with stride 2, $\boldsymbol{A}_{41}$ is a $(20 \cdot 32 \cdot 32) \times 100$ dense linear layer, $\boldsymbol{A}_{42}$ is a $(60 \cdot 16 \cdot 16) \times 100$ dense linear layer $\boldsymbol{A}_{43}$ is a $(80 \cdot 8 \cdot 8) \times 100$ dense linear layer, and $\boldsymbol{A}_{44}$ is a $100 \times 100$ dense linear layer. The corresponding variable $\boldsymbol{q}$ as in Equation (9) then has 4 elements of shape $(30 \times 32 \times 32), (60 \times 16 \times 16), (80 \times 8 \times 8), (100 \times 10)$. When applying the proximal operator to $\boldsymbol{q}$, we use $1, 6, 8, 10$ as their number of groups, respectively.

**mDBM Training details and hyperparameters** Treating the image reconstruction as a dense classification task, we use cross-entropy loss and class weights $\frac{1-\beta}{1-\beta^{n_i}}$ with $\beta = 0.9999$ (Cui et al., 2019), where $n_i$ is the number of times pixels with intensity $i$ appear in the hidden pixels. For classification, we use standard cross-entropy loss. To enable joint training, we put equal weight of 0.5 on both task losses and backpropagate through their sum. For both tasks, we put $\tau_i \boldsymbol{\Phi} \boldsymbol{q}_i^*$ into the cross-entropy loss as logits, as described in Equation (17). Since mean-field approximation is (conditionally) unimodal, the scaling grants us the ability to model more extreme distributions. To achieve faster damped forward-backward iteration, we implement Anderson acceleration (Walker & Ni, 2011), and stop the fixed point update as soon as the relative difference between two iterations (that is, $\|\boldsymbol{q}_{t+1} - \boldsymbol{q}_t\|/\|\boldsymbol{q}_t\|$) is less than 0.01, unless we hit a maximum number of 50 allowed iterations. For $\text{prox}_f^\alpha$ and the damped iteration, we set $\alpha = 0.125$ (Although one can tune down $\alpha$ whenever the iterations do not converge, empirically this never happens on our task).

We use the Adam optimizer with learning rate 0.001. For MNIST, we train for 40 epochs. For CIFAR-10, we train for 100 epochs using standard data augmentation; during the first 10 epochs, the weight on the reconstruction loss is ramped up from 0.0 to 0.5 and the weight on the classification loss ramped down from 1.0 to 0.5; also during the first 20 epochs, the percentage of observation pixels is ramped down from 100% to 50%.

**Deep RBM Training details and hyperparameters** The deep RBM is trained using $CD$-1 algorithm for 100 epochs with a batch size of 128 and learning rate of 0.01.

**Convergence of inference during training** We note that, comparing to the differently-parameterized monDEQ in Winston & Kolter (2020), whose linear module suffers from drastically increasing condition number (hence in later epochs taking around 20 steps to converge, even with tuned $\alpha$), our parameterization produces a much nicer convergence pattern: the average number of forward iterations over the 40 training epochs is less than 6 steps, see Figure 7.

**mDBM patch imputation experiments** We train mDBM on the task of MNIST patch imputation. We randomly mask a $14 \times 14$ patch, chosen differently for every image, similar to the query training in Lázaro-Gredilla et al. (2020). To make the model class richer, we lift the monotonicity constraint, and find that the model converges regardless. Our model reconstructs readable digits despite potentially large chunk of missing pixels (Figure 8b). If the model is given the image labels as input injections, our model

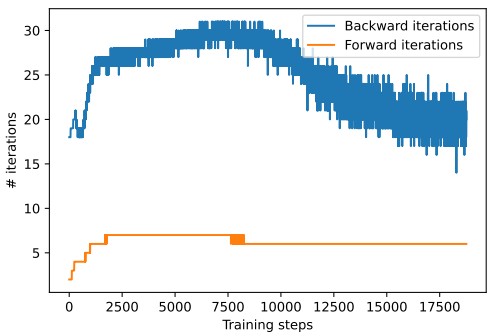

Figure 7: Convergence of forward-backward splitting.

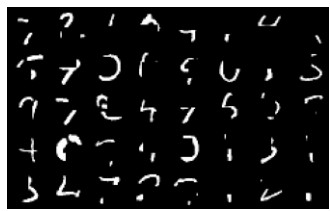

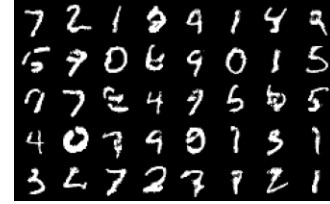

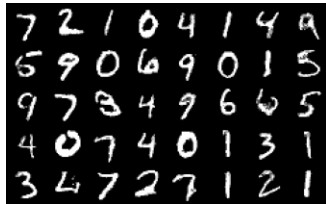

(a) Test data with a $14 \times 14$ patch masked

(b) Imputation with a $14 \times 14$ patch masked, inference without injection labels

(c) Imputation with a $14 \times 14$ patch masked, inference with injection labels

Figure 8: MNIST pixel patch imputation using mDBM

performs conditionaly generation fairly well (Figure 8c). These results demonstrate the flexibility of our parameterization for modelling different conditional distributions.

**Deep RBM results using Gibbs sampling**   The deep RBM is trained as before. For joint imputation and classification, the DBM uses Gibbs sampling of 10000 and 100 steps respectively, although the quality of the imputed image and test accuracy are insensitive to the number of steps.

We randomly mask off 60% pixels, or a randomly selected $14 \times 14$ patch; the results are shown in Figure 9, and are better than when mean-field inference is used, (shown in Figure 3).

In the experiment with 60% of pixels randomly masked, we also test the model on predicting the actual digit simultaneously. The test accuracy is 93.58%, comparable to the mDBM accuracy of 92.95%.

**Comparison of inference methods**   We conduct numerical experiments to compare our inference updating method to the ones proposed by Krähenbühl & Koltun (2013); Baqué et al. (2016), denoted as Krähenbühl's and Baqué's respectively. Krähenbühl's fast concave-convex procedure (CCCP) essentially decomposes to Equation (11), the un-damped mean-field update with softmax. This update only converges provably when $\mathbf{\Phi}$ is concave. Baqué's inference method can be written as

$$\boldsymbol{q}_h^{(t)} = \text{softmax}\left((1 - \alpha)\log\boldsymbol{q}_h^{(t-1)} + \alpha(\mathbf{\Phi}_{hh}\boldsymbol{q}_h^{(t-1)} + \mathbf{\Phi}_{ho}\boldsymbol{x}_o + \boldsymbol{b}_h)\right). \tag{20}$$

This algorithm provably converges despite the property of the pairwise kernel function. However, this procedure converges in the sense that the variational free energy keeps decreasing. Therefore their fixed point may not be the true mean-field distribution Equation (7). In this experiment, we train the models using three different updating methods, and perform inference using three methods as well, with and without the monotonicity condition.

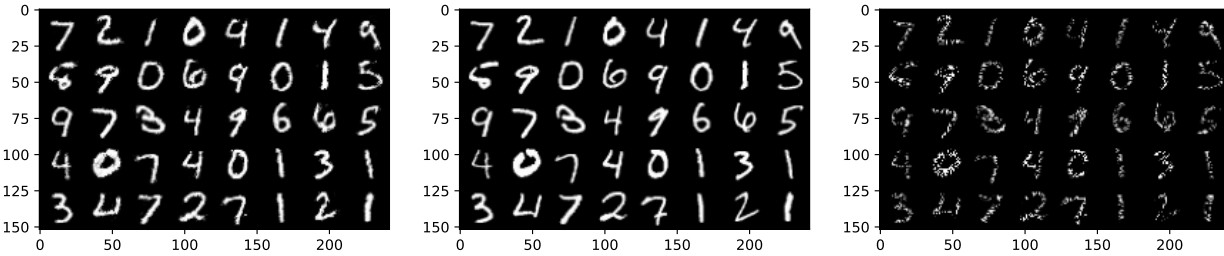

(a) 60% pixels are randomly masked. From left to right: imputed image, true image, masked image.

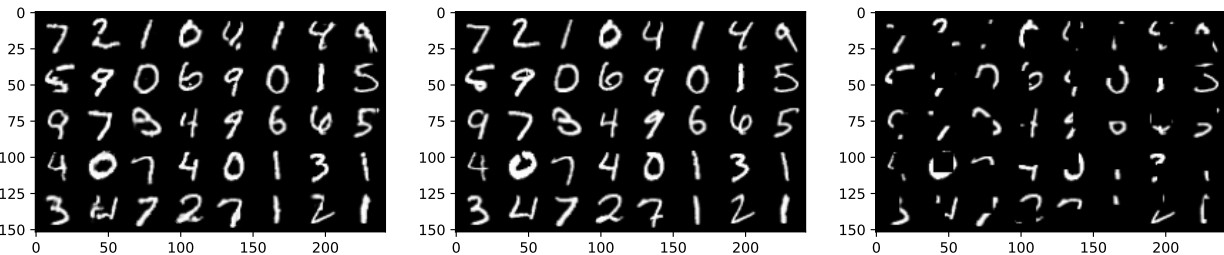

(b) $14 \times 14$ patches are randomly masked. From left to right: imputed image, true image, masked image.

Figure 9: RBM for image imputation using Gibbs sampling

We also compare the convergence of our method to the regularized Frank-Wolfe method in Lê-Huu & Alahari (2021). Their update step can be written as

$$\boldsymbol{q}_h^{(t+1)} = (1 - \alpha)\boldsymbol{q}_h^{(t)} + \alpha \operatorname{softmax}\left(\frac{1}{\lambda}(\boldsymbol{\Phi}_{hh}\boldsymbol{q}_h^{(t)} + \boldsymbol{\Phi}_{ho}\boldsymbol{x}_o + \boldsymbol{b}_h)\right).$$

We use $\lambda = 0.7$ as in their paper. Our method converges faster than the FW method. See the result in Figure 10.

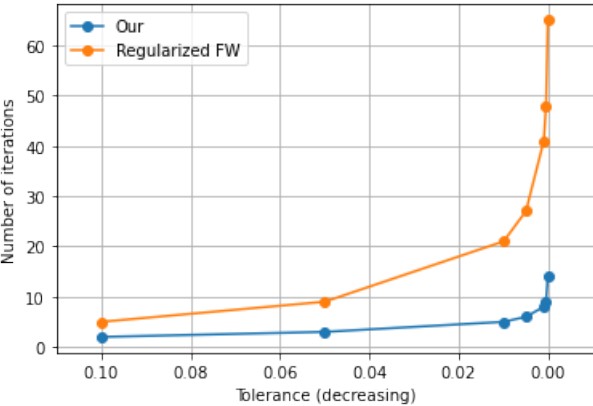

Figure 10: Convergence of our method vs. the regularized FW. This experiment is done using the same setup as in Figure 11.

Krähenbühl's and Baqué's methods often do not converge in the backward pass (there's no theoretical guarantees neither). To rule out the impact of the backward iteration, during training we directly update use the gradient of the forward pass, instead of using a backward gradient hook to compute Equation (16). Figure 12 and Figure 13 demonstrate how the three update methods impute missing pixels when trained with different update rules, with and without the monotonicity condition, respectively. Krähenbühl's usually does

Table 4: Relative update residual when monotonicity is not enforced

| Inference / Train | Krähenbühl | Baqué | Our |
|---|---|---|---|
| Krähenbühl | 0.0005 | 0.0065 | 0.0024 |
| Baqué | 1.0924 | 0.0119 | 0.0042 |
| mDBM | 1.1286 | 0.0065 | 0.0022 |

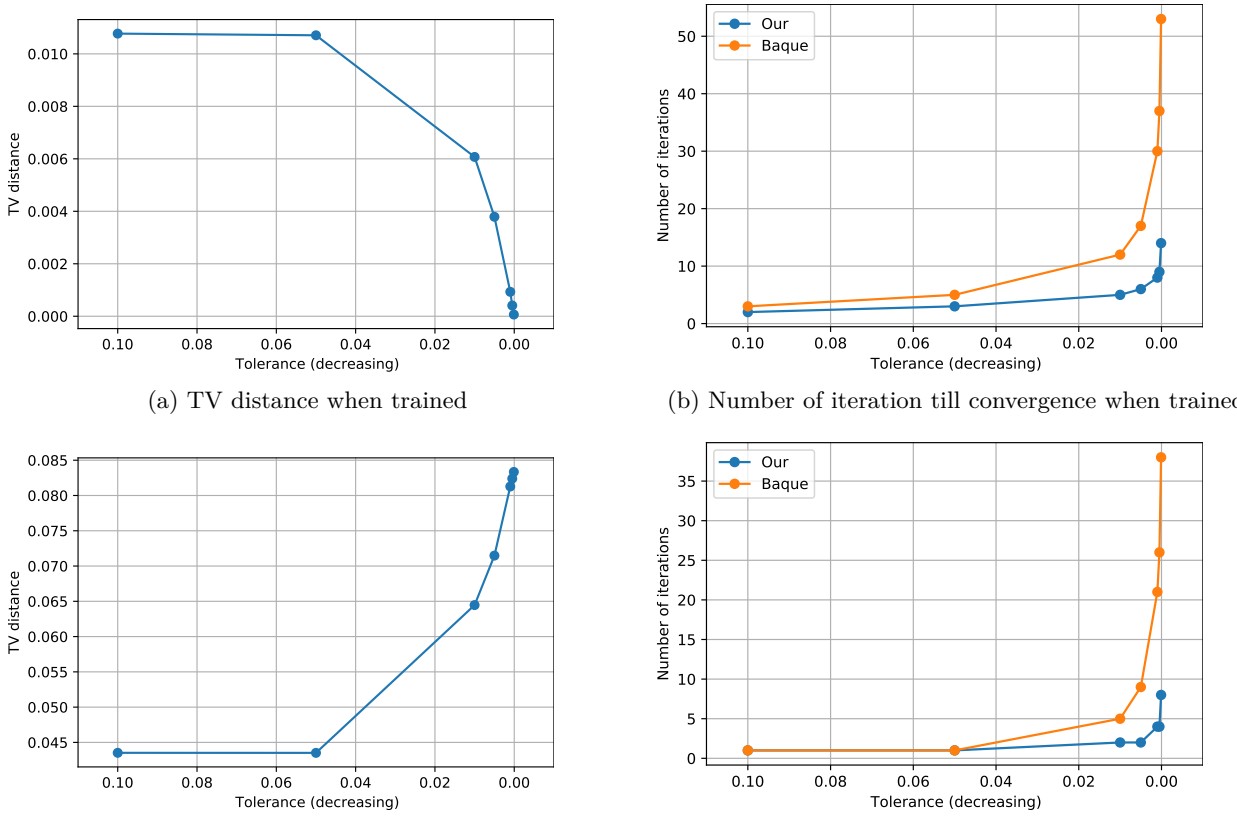

(a) TV distance when trained

(b) Number of iteration till convergence when trained

(c) TV distance at initialization

(d) Number of iteration till convergence at initialization

Figure 11: TV distance and convergence speed

not converge when the model is trained with our method or Baqué's, whereas the other two methods impute the missing pixels well. The classification results are presented in Table 5 and Table 6. Notice that when trained with our method or Baqué's, the convergence issue of Krähenbühl's leads to horrible classification accuracy. Our method is superior to other inference methods when the model is trained in a different update fashion. For example, if the model is trained by using Krähenbühl's, it makes sense that the model performs the best if the inference is also Krähenbühl's since the parameters are biased toward that particular inference method. However, our method in this case outperforms Baqué's.

After these methods halt and return $\boldsymbol{q}_h^T$, we run one more iteration of

$$\boldsymbol{q}_h^{T+1} = \text{softmax}\left(\boldsymbol{\Phi}_{hh}\boldsymbol{q}_h^T + \boldsymbol{\Phi}_{ho}\mathbf{x}_o + \boldsymbol{b}_h\right), \tag{21}$$

and record the relative update residual $\|\boldsymbol{q}_h^{T+1} - \boldsymbol{q}_h^T\|/\|\boldsymbol{q}_h^T\|$ for randomly selected 4000 MNIST images. The results are listed in Table 3 and Table 4. To alleviate the effect of numerical issues, we strength the convergence condition to either the relative residual is less than $10^{-3}$ or the number of iterations exceeds 100 steps.

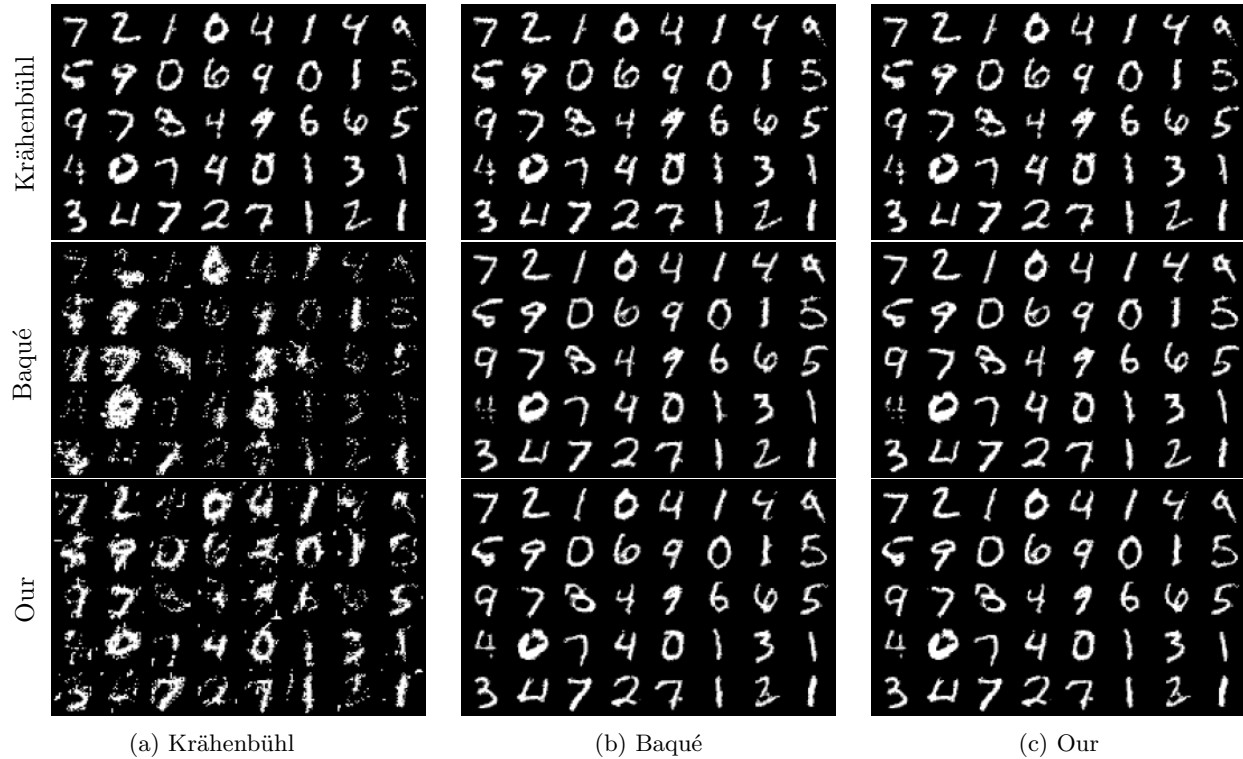

Figure 12: Training and inference using all three update rules with 40% observed pixels *with* the monotonicity condition. The labels on each row represent the training update rule, and the labels on the columns represent the inference update rule.

Table 5: Classification error (standard deviation) when monotonicity is enforced

| Inference / Train | Krähenbühl | Baqué | Our |
|---|---|---|---|
| Krähenbühl | **0.042 (0.0013)** | 0.114 (0.0019) | 0.0498 (0.0014) |
| Baqué | 0.958 (0.0013) | 0.038 (0.0010) | **0.034 (0.0012)** |
| mDBM | 0.946 (0.0024) | 0.0425 (0.0016) | **0.0412 (0.0017)** |

It appears in Table 3 and Table 4 that although our method has a much lower residual compare to Baqué's, both of them seem small and convergent. This is because the "optimal" fixed point in this setting on MNIST might be unique and both methods happen to converge to the same point. However, this is in general not true. We compare our method vs Baqué's on 400 randomly selected MNIST test images with 40% pixels observed, and perform mean-field update until the relative residual of $[0.1, 0.05, 0.01, 0.005, 0.001, 0.0005, 0.0001]$ is reached (without step constraint), respectively. Then we measure the TV distance between the distributions computed by these two methods on the remaining 60% pixels, as well as the convergence speed. The results are demonstrated in Figure 11. One can see that when the model is trained (using Krähenbühl's, Figure 11a), the TV distance converges to 0 as the tolerance decreases. However, when the model is just initialized (but still constrained to be monotone), the TV distance remains large (Figure 11c). Even though in this case the optimal fixed point may be unique, our method is still superior to Baqué's: it takes us less iterations till convergence, despite whether the model is trained or not.

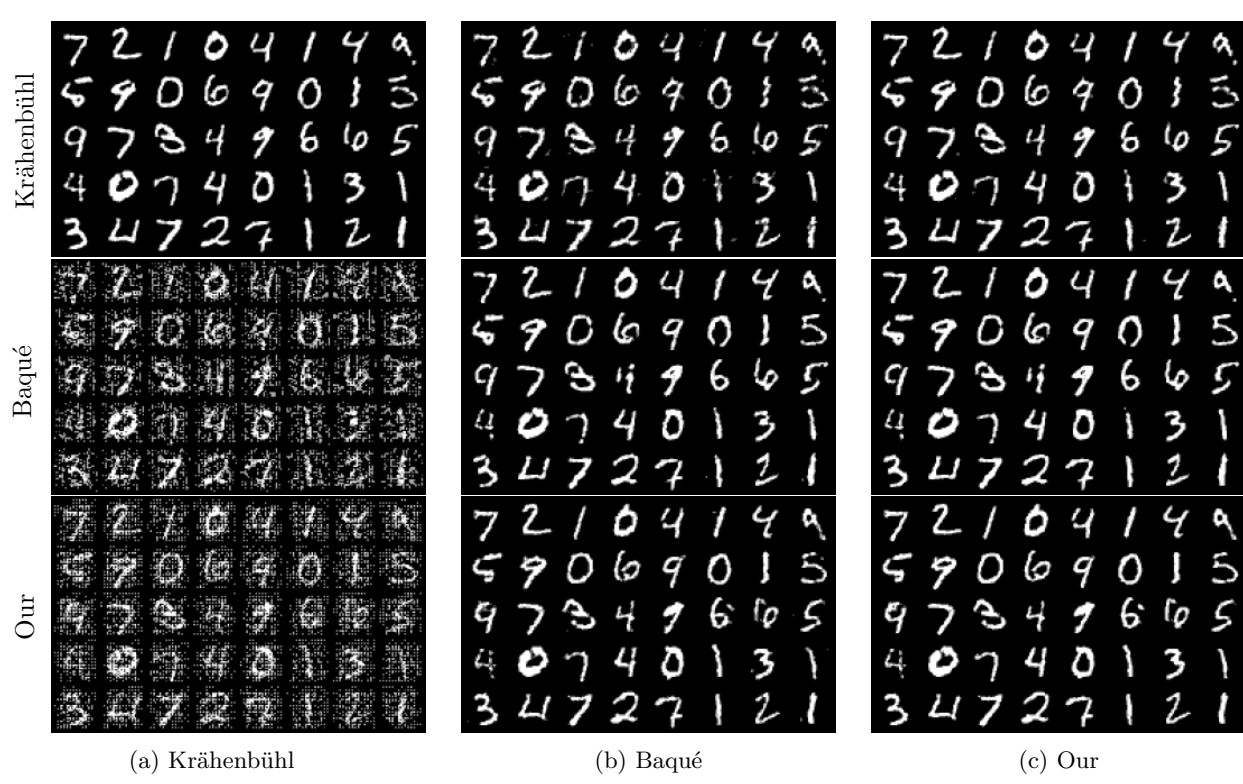

(a) Krähenbühl        (b) Baqué        (c) Our

Figure 13: Training and inference using all three update rules with 40% observed pixels *without* the monotonicity condition. The labels on each row represent the training update rule, and the labels on the columns represent the inference update rule.

Table 6: Classification error (standard deviation) when monotonicity is not enforced

| Train \ Inference | Krähenbühl | Baqué | Our |
|---|---|---|---|
| Krähenbühl | **0.035 (0.0017)** | 0.189 (0.0023) | 0.051 (0.0015) |
| Baqué | 0.762 (0.0013) | **0.041 (0.0013)** | 0.055 (0.0012) |
| mDBM | 0.90 (0.0002) | 0.063 (0.0021) | **0.036 (0.0017)** |

