# OpenReview forum: "Monotone deep Boltzmann machines"
_TMLR — Accepted by TMLR_

### Review · Reviewer_iVQk · 2023-01-27

**Summary Of Contributions:**

The authors take inspiration of recent work on DEQs allowing for provable convergence and extend these ideas to deep Boltzmann machines. They provide a restricted set of deep Boltzmann machines that crucially allows for recurrent connections. They then provide theory allowing for provable convergence of their mean-field updates and practical insights into how to enable fast inference with their model. The theory is accompanied with classic experiments on vision datasets (MNIST, CIFAR10).

**Audience:**

Yes

**Broader Impact Concerns:**

-

**Claims And Evidence:**

Yes

**Requested Changes:**

I have no requested changes although a couple of suggestions.
It would be nice to get a bit better feeling of the algorithm and how it compares to related work:

How fast/slow is the algorithm (and the compared methods) in wall-clock time? It would be nice how long the different components scale with network / dataset size. It would be useful to provide pseudocode for training and inference to allow for an overview of the different algorithmic components.
In general, the paper is very much concerned about provable convergence. As mentioned by the authors (and in related work), quite surprisingly at least for me,  the dynamical systems described by the DEQ / your updates seem often to converge without additional regularisation or more elaborate intervention. Could you maybe elaborate on that? I think this part falls a bit short in the paper.


**Strengths And Weaknesses:**

The paper is well written and presents the ideas and novelties (and shortcomings, connections to related work) in a transparent manner.  Although I did not check the theoretical claims thoroughly, in that respect the paper is quite dense and requires quite a lot of background knowledge, the presentation of the results is concise and seems correct.
Arguable the theoretical novelty is limited but I still like the new connection to DBMs, and the novel theoretical results that allow for this bridge. I think that the paper is well executed and will be well received and appreciated.

---

> ### Author Response · Authors · 2023-02-14
> **Thanks for your review**
>
> ### Wall-clock time:
>
> We found it quite hard to compare using wall-clock time since we have many components that run till convergence: during forward iteration, we need to compute eq (12) iteratively till convergence; at each iteration, solving the KKT condition in theorem 3.2 is also an iterative algorithm. The backward iteration usually takes longer than the forward iteration as shown in figure 7. These amount to a large variance in the wall-clock time. Typically the model we use in the paper can take between 2~7 minutes (with one A6000) training on 1 epoch of all MNIST images. As a comparison, training RBM with CD-1 takes less than 5s with one epoch of MNIST (of course increasing the number of Gibbs steps increases training time but CD-1 is what people use mostly). This really is one of the shortcomings to all types of DEQ models. However, inference is a lot faster. For the RBM baseline, it takes 4s to fill 60% pixels given 40\%, running 10000 mean-field inference steps. The non-monotone DEQ takes ~0.82s to converge and the monotone version takes ~0.3s to converge. Here the models take a batch of size 40 images.
>
> ### Pseudocode
>
> It can be found at the end of section 3 in the revision.
>
> ### Convergence of non-guaranteed methods
> Just to clarify, to give provable convergence, we need to enforce $I-\Phi\geq mI$ for some $m>0$. This does require careful "regularization". In this work, we parameterize the model so that it always satisfies the condition, so no explicit regularization is needed.
>
> We also did experiments with a "non-monotone" version, where the model is still parameterized as in eq(3), except that we didn't project $A$ onto a norm ball (as in eq (5)). In this case, convergence really depends on the task. We notice that for simpler imputations like missing random pixels, the method without regularization converges fine. For harder tasks like filling a continuous block of pixels, the non-convergence algorithm often doesn't converge within the max number of steps (in our experiments the maximum number of steps is 50, and the tolerance is $10^{-2}$), especially the backward iteration. However, even if the algorithm doesn't converge to the tolerance, the gradient computed is benign enough that it still decreases the training loss. This is an interesting point about whether we can still have good empirical convergence when $m$ is negative but has a small absolute value, so that we can control the level of "non-monotonicity". It is mentioned in the future direction.

---

### Review · Reviewer_ZkY7 · 2023-01-31

**Summary Of Contributions:**

Boltzmann machines are a classical fully-connected model in which computing (properties of, or sampling from) stationary distributions is intractable.
Restricted Boltzmann machines and their deep counterparts wherein connections are organised in a layered structure allow for tractable inference via a form of Gibb's sampling called contrastive divergence.
The authors consider another type of connectivity structure that allows for tractable inference.
They call the structure the monotone DBM, where arbitrary self-connections are allowed in each layer but the weights are restricted in such a way as to allow a unique fixed point.
This is achieved through a connection to recently proposed implicit layers called deep equilibrium models, and in particular, monotone deep equilibrium models with special activation functions.
Monotone DBMs are applied to joint image completion and classification tasks.

**Audience:**

Yes

**Claims And Evidence:**

Yes

**Requested Changes:**

**Minor recommendations:**
- Start of section 2. "Deep equilibrium models, especially their convergent version, the monotone". The phrase "their convergent version" seems to imply there is a singular convergent version, but there exist multiple. For example, [1-4] and others that are recently published.
- Theorem 3. The first part of the theorem 1) $\Phi_{ii} = 0$ (which I believe should be a matrix of zeros) is trivial, by (4). Why include it in the theorem statement?
- First equation in section 3.2. Is $E$ the set of all edges? Is $o$ the set of observed nodes?
- I find the notation where the fixed point of a DEQ $z(x)$ is an explicit function of $x$, but then later in for example equation (15) it is not. I suggest picking one notation and keeping it consistent.
- Text above equation (17). ``owning'' should be ``owing''.
- Figure 3. Why are (a) and (d) different figures? It seems like a different seed of a random mask was applied to each. And I believe (c) and (f) are the same. Can you make this figure where there is a single figure instead of (a) and (d), and a single figure instead of (c) and (f)?
- Why did you compare with mean-field updates on the RBM, if Gibbs sampling/contrastive divergence is the de facto standard? Perhaps move the result from the appendix to the main text?


[1] Ezra Winston and J Zico Kolter. Monotone operator equilibrium networks. Advances in Neural Information Processing Systems, 33:10718–10728, 2020.
[2] Max Revay, Ruigang Wang, and Ian R Manchester. Lipschitz bounded equilibrium networks. arXiv preprint arXiv:2010.01732, 2020.
[3] Laurent El Ghaoui, Fangda Gu, Bertrand Travacca, Armin Askari, and Alicia Tsai. Implicit deep learning. SIAM Journal on Mathematics of Data Science, 3(3):930–958, 2021.
[4] Tsuchida, R., & Ong, C. S. (2022). Deep equilibrium models as estimators for continuous latent variables. arXiv preprint arXiv:2211.05943.

**Strengths And Weaknesses:**

**Strengths:**
- The question of whether more general Boltzmann Machines than RBMs admit efficient inference procedures is an interesting one. This paper answers that question in the positive, although perhaps in a somewhat restricted setting. It leverages other parts of the literature, namely Deep Equilibrium Models, in order to do so, which I think is nice.
- The paper is mostly well written. Excluding isolated instances, it is easy to follow the train of thought and notations of the authors.

**Weaknesses:**
- Section 3.3 does not clearly describe the conversion of ``neural network convolution operators'' to parameterisation in terms of $A$. I do appreciate that this is a difficult thing to convey, due to the various subscripts involved. Also the mention of the spectral normalisation of the neural net style convolution operator. I encourage the authors to spend some time thinking about the best possible way to communicate this, because if this is done well, it will be very useful for other researchers. Deferring the explanation to the Appendix is suboptimal.
- As the authors admit in the abstract, the work is largely at a conceptual stage. The datasets and problems considered are largely toy from a ``deep learning perspective'', but not from a `graphical model' perspective. I think it is good that the authors acknowledged this weakness in the abstract, experimental evaluation, and somewhere else towards the beginning, and I do not see this as a substantial enough weakness to prevent publication at TMLR.

**Questions** (for clarification --- not necessarily strengths or weaknesses):
- First paragraph. Are DBMs necessarily discrete? I seem to recall some versions allow for continuous-valued nodes.
- Figure 1, right. Why did you choose to draw a general BM in this way? It appears as though what you have drawn is not a complete graph, unless I misunderstand the notation or terminology. For example, the top right node is not connected to the top middle node. I am more familiar with the layout where the nodes are arranged on a circle, and every node is connected to every other node.
- Equation (4). What is $\Phi_{ij}$? The matrix $A_i$ is a $d \times k_i$ matrix, so I guess $\Phi_{ij}$ should be a $k_i \times k_j$ matrix? So then the ``0'' listed in the second line of equation (4) is actually $0_{k_i \times k_j}$? By the way, I think there is a minor typo in the text two lines after equation (1). You write $\Phi_{i,j}$ but I think it should be $\Phi_{ij}$.
- The monotone DEQ construction is guaranteed to admit a unique fixed point under a certain condition like $I - \Phi \succ m I$, which translates to a certain condition on $A$ or $A$ hat. I believe these conditions are easy to impose by always enforcing the spectral constraint and then running any gradient based optimiser. Is that true? Or does the gradient based optimiser have to explicitly enforce a condition on the parameters it is learning?
- Perhaps some of the works below and the others that you can find might be useful for your aspiration (4) in Future directions. Namely, other activation functions can be obtained using other frameworks that still allow for the existence of a unique fixed point. Some of these also admit probabilistic interpretations.

All in all, I am leaning to accept this paper. I think some of the audience of TMLR will find it very interesting, and I find the claims to be without any serious issues.

---

> ### Author Response · Authors · 2023-02-14
> **Thanks for your review**
>
> ### Description of of convolution parameterization in terms of $A$
>
> Thank you for the feedback. The revision now has some explanation in the main text.
>
> ### Response to questions
>
>  * Good point, classical DBMs are discrete but there are continuous versions as well. Here we only consider the classical DBMs.
>
>  * You are correct, the BM should be a complete graph. The red edges were meant to represent examples of edges which exist in the general BM but not the RDBM. We will clarify this.
>
>  * Clarifying eq (4): your understanding of the matrix is correct, $\Phi_{ij}$ is indeed $k_i\times k_j$, the "0" is technically $k_i\times k_j$ but since the case is $i=j$ so they align. We have fixed the typo.
>
>  * The spectral condition on $A$ is enforced by normalizing $A$ by the desired norm at every iteration (after taking a full grad update). Because $\hat{A}$ is spectrally-normalized as in eq (2), the condition $I-\Phi \succ mI$ is guaranteed by construction (see Theorem 3.1) and can be trained using unconstrained gradient-based optimization.
>
>
>  * Thanks very much for your suggestions regarding future work (4). We put them in the background section and the future direction section in the revision.
>
> ### Response to requested changes
>
> Thank you for your recommended edits. We have updated them in the review. Here are some additional explanations:
>
>  * Theorem 3: The statement follows immediately but we want to make the theorem self-contained.
>
>  * E and o: your understanding is correct.
>
>  * Figure 3: The experiments are now using the same random seeds in the revision.
>
>  * Why compared with mean-field updates?: The reason is that mDBM essentially models the meanfield inference process of Boltzmann machines. One can use mDBM to sample in an autoregressive fashion but we cannot model the convergence of Gibbs sampling as the convergence of a DEQ. It is an apple-to-apple comparison to RBM mean-field inference.

---

> > ### Comment · Reviewer_ZkY7 · 2023-02-16
> > **Thanks for your updates**
> >
> > Thanks for responding to my review.
> >
> > - Theorem 3. Fair enough. Agree that a self-contained statement has value.
> > - Understood about the mean-field updates.
> >
> > I think some of the audience of TMLR will find this paper very interesting, and I find the claims to be without any serious issues. I will wait and see if the other reviewers are still concerned about additional experimental benchmarks after they respond to the rebuttal.

---

### Review · Reviewer_djWs · 2023-02-06

**Summary Of Contributions:**

The Authors present a new approach for deep Boltzmann machines, which leverages recent advances on equilibrium models to propose a novel inference scheme for DBMs. This is achieved by applying the results of Baie et al. (2019) and Winston & Kolter (2020) regarding fixed-point convergence of infinitely deep neural networks with input injection (of the class $\sigma(Wz+Ux+b)$) to DBMs, allowing to derive a scalable variational inference scheme. This involves constraining the linear model to be compatible with the DBM architecture (i.e. graphical model with no-loops) and to ensure the requirement for Winston & Kolter (2020) to hold (i.e. monotonicity). After an in-depth explanation of the proposed method, the Authors conclude the paper with a small section with experiments on MNIST and CIFAR datasets, mostly on image imputation and marginally on image classification.

**Audience:**

Yes

**Claims And Evidence:**

Yes

**Requested Changes:**

- Evidence of convergence to fixed point: beyond the theoretical guarantees, would it be possible to show the fixed point convergence empirically?
- Effect of temperature: you quickly mentioned some temperature scaling to the output marginals but you didn't, as far as the paper goes, tested different configurations. Didn't you find some pathological situations where this was necessary? What would happen with different setups?
- Summary of the method and algorithm: the developed method involves several steps and different parameterizations and approximations; it would be nice if you could provide a paragraph at the end of section 3 to summarize the proposed method, maybe with the aid of a pseudo-algorithms.
- Additional experimental evaluation: the paper for the moment has a limited experimental campaign, based only on 2 datasets and 1 comparison (the naïve RBM). One possibility is to benchmark mDBM with other generative models for e.g. image imputation (see 1-4 for some ideas). To be clear, for me the paper has value beyond the ranking on this benchmark, but I would encourage the Authors to explore with different models. This would be also a good opportunity to briefly discuss and compare computational aspects of your proposal.


(1) GAIN: Missing data imputation using generative adversarial nets. ICML 2018

(2) MisGAN: learning from incomplete data with generative adversarial networks. ICLR 2019

(3) MIWAE: Deep generative modelling and imputation of incomplete data sets. ICML 2019

(4) MCFlow: Monte Carlo Flow Models for Data Imputation. CVPR 2020

**Strengths And Weaknesses:**

**Strengths**:
- The paper is overall well written paper, relatively easy to follow.
- Elegant formulation and nice application of recent advances on deep learning methods to more classic models, like DBMs
- Some implementation details needed to apply the results in Winston & Kolter (2020) are carefully explained, providing from a methodological point of view a nice contribution

**Weaknesses**
- Limited experimental evaluation (see below)

---

> ### Author Response · Authors · 2023-02-14
> **Thanks for your review**
>
> ### Evidence of convergence
>
> In the main text experiment section we include some results in "Comparison of inference method", where we show the convergence of several methods on MNIST. More details can be found in appendix B. The convergence during training is presented in figure 7 (in the revision).
>
> ### Effect of temperature
>
> There is this pathological behavior that without the temperature, cross-entropy loss tends to drive the model to an output distribution that is too uniform. Although in this case, we didn't really train the model for too many epochs, so it is possible that it's just taking a longer time to converge. Note that the temperature only helps with optimization, it is not used during inference. The temperature also does not affect the model's (probabilistic) expressiveness in the argmax sense, it only makes correct predictions "more correct". Here's a concrete example: let's say we predict the pixel in a fixed position, this is the $q_i(x_i)$.  We bin the model into 4 categories, so $x_i\in[0,1,2,3]$, and for each of them the model outputs a probability, and the prediction is the argmax. The temperature scaling $q_i^{\tau_i}$ will *not* make wrong predictions correct, it only provides a desirable implicit bias by reducing distribution entropy.
>
> ### Summary of method and algorithm
>
> Thank you for the suggestion. This is now included in the revision.
>
> ### Additional experimental benchmarks
>
> We want to make the paper emphasizes this proof of concept that there's a tight connection between mean-field inference and DEQs, instead of a state-of-the-art algorithm. We consider this type of benchmark beyond the scope of our paper, and it may potentially make our delivery less concentrated. It is however a good effort to benchmark mDBM and we have added the references you mentioned to the "future work" section.

---

### Review · Reviewer_zuLa · 2023-02-08

**Summary Of Contributions:**

Sorry for the late review here.

This paper proposes monotone deep boltzmann machines (mDBMs), which are a restriction of deep Boltzmann machines to the class of graphs that have no self-loops within a layer, but can connect both forwards and backwards in the layers. mDBMs have the favorable property that they can be parameterized as a deep equilibrium model (DEQ), which gives reasonably efficient training procedures and global solutions.

Experiments are performed in MNIST and CIFAR-10 in an unsupervised fashion, demonstrating that this inference procedure is quite flexible and reasonably accurate.

**Audience:**

Yes

**Claims And Evidence:**

Yes

**Requested Changes:**

Requested experiments

- My understanding is that DEQ based models (e.g. https://arxiv.org/pdf/2006.08591.pdf)  and some of the DBM results (e.g. https://www.scinapse.io/papers/2100495367) tend to have experiments in classification or regression using the DBM as a feature constructor. If this is possible with the mDBM, could this be done here? That is, classifying cifar-10 classes based on the states from the mDBM, not just in-filling.

- Figure 4: what is the extrapolation performance of the mDBM on MNIST and CIfAR-10? For example, mask the left 50% of pixels and then attempt to generate the right half.

These are mostly presentation based comments:

- pg 4: redefine phi, I believe it's the connection network of the graph?

- pg 5: "We see that q_h is the ...": explain this further as it seems both important and not obvious at first glance

- pg 5: where is the proof of proposition 3.1? mark this in the main text

- eq 8: use \mathbb{I} instead of I

- pg 6: "height x ... x " use $x$ for the x-es here

- pg 7: how are m and L determined prior to solving the system? I think m might be mentioned above, but L is not

- pg 7: give a sentence describing Halley's method

- pg 7: "extend their work to the softmax nonlinear operator": I thought that your work used the proximal operator instead of the softmax one?

- pg 8: "This backward pass can also be computed via an iterative approach..." explain what their approach is

- tables 1,2: Why do the errors go down as the masking percentage goes up?

- tables 1,2: please replace "our" with mDBM

- table 3: What does relative residual update mean in this context? I'm also more generally confused by the "comparison of inference methods" section. Is the conclusion that your method and Baque's methods are much more globally accurate?

- pg 11: "although we have a fairly efficient implementation ..." why not one step newton approach or construct a polynomial/ taylor series  approximation? It seems like there are well-known approximation methods for the Lambert W function, like [this one](https://nvlpubs.nist.gov/nistpubs/jres/65B/jresv65Bn4p245_A1b.pdf).

**Strengths And Weaknesses:**

Strengths:

- The proposed method seems to be technically sound and is well explained mathematically. It also draws from a much newer, powerful class of DEQ models that seem to work well in practice.

- I like the idea of constructing a highly structured parameterization to ensure that the necessary conditions are satisfied immediately.

Weaknesses:

- The experiments seem to be quite limited, especially compared to the current powerful methods of diffusion models in generative modelling and even DEQ models, which can be used for sequence modelling as well. I guess I'm asking what can a (m)DBM do that these approaches cannot?

- There's no clear experiments on convolutional nets (I could be missing them?) although that is listed as a selling point of the approach.

---

> ### Author Response · Authors · 2023-02-14
> **Thanks for your review**
>
> ### What can a (m)DBM do that generative diffusion models and DEQ models cannot?
>
> A vanilla DEQ model cannot model random variables. A diffusion model is as capable as mDBM in terms of joint inference [1]. There is another line of works that focus on combining diffusion models and DEQs, but in this paper we want to focus on exploring the connection between RBMs and DEQs. To be candid, a diffusion model will almost certainly outperform mDBM (or any kind on RBM). We want to emphasize that this paper is presented as a proof of concept, rather than a state-of-the-art algorithm.
>
> ### No clear experiments on convolutional nets
>
> The existing experiments are using small CNNs, as described in appendix B.1. We will move this detail to the main text. Apologies for the confusion.
>
> ### Experiments in classification or regression using the DBM as a feature constructor
>
> While it is conceptually possible to use the mDBM as a feature constructor, we believe it is beyond the scope of this paper. Here we have focused on the unique properties of mDBM as an end-to-end probabilistic model, which has the ability to perform classification and imputation jointly. To be precise, both mDBM and the RBM in our paper are already tested for classification, where the label is treated as a random variable in the inference process.
>
> ### Extrapolation performance of mDBM
> We add extrapolation experiments (masking lower half of image) in the revision (see figure 5). This is a harder task but we still observe reasonable extrapolation on MINST and nontrivial extrapolation on CIFAR10.
>
> ### Presentation-based comments:
>
>  Thank you for your presentation suggestions, which we will incorporate (see updated pdf). To answer your specific questions:
>  * $q_h$: this is the vectorization of the previous equation.
>  * Proof of Prop. 3.1: The proof follows the statement of the proposition, but it was not wrapped in a "proof" environment. Fixed now.
>  * pg 6 height x.. x: these are multiplications, so $x$ is not the desired form?
>  * $m$ is a design choice enforced by the parameterization described in Section 3.1. The Lipschitz constant $L$ of $I-\Phi$ is it's spectral norm, but isn't actually needed (other than for bounding the convergence rate).
>  * softmax vs proximal operator: We use the softmax operator in the sense that the we are finding the mean-field fixed-point given in (6), corresponding to softmax. To _compute_ the fixed-point, however, we need the fact that softmax corresponds to a proximal operator $prox_f^1$. To achieve convergence we use the dampened proximal operator $prox_f^\alpha$.
>  * Iterative approach for backward pass: Computing the gradient in (16) is cast as another operator splitting problem and solved using, e.g. forward-backward splitting. Further description is now added to the revision.
> * Error trend in tables 1,2: the wording ``masked'' here is confusing, it means 20\%, 40\% ... is shown to the classifier.
> * Relative update residual: it means $||q_h^{t+1}-q_h^{t}||/||q_h^{t}||$, this is added to the main text. The two other methods are known heuristics that compute the fixed point of softmax. We show that they either diverge in some cases or don't converge as well as ours.
> * One step Newton or other approximation of Lambert W function: We use Halley's method instead of Newton's method as it is the preferred manner [2]; and even if we use other types of approximation, the bottlenecks here are a) solving the KKT condition in theorem 3.2, and b) implicit differentiation as described in section A.3. These will be slower than directly computing ReLU/softmax anyway.
>
> [1]: [CARD: Classification and Regression Diffusion Models](https://arxiv.org/abs/2206.07275)
>
> [2]: [On the Lambert W Function](https://cs.uwaterloo.ca/research/tr/1993/03/W.pdf)

---

### Decision · Action_Editors · 2023-03-22

**Recommendation:** Accept as is

**Comment:**

This is generally a strong paper with some interesting ideas that I believe is highly suitable for publication at TMLR.  Though reviewers originally raised some concerns about experimental comparisons, these were all things that were either adequately addressed in the rebuttal or that the reviewers did not feel should prohibit acceptance.  As the authors have already made updates to the paper based on the reviewers' comments and there are no significant issues still outstanding, I recommend accepting the paper "as is".

**Audience:**

The paper will be of clear interest to the TMLR audience: its research is within the scope of the journal and the paper contains novel and non-trivial new material.

**Claims And Evidence:**

All the reviewers and I are satisfied that the claims the paper makes are adequately evidenced.  Though the paper does not have especially extensive experiments or impressive numerical results, the claims made are appropriately modest and there are no unjustified conclusions; the authors are generally quite forthcoming with the limitations of the approach.  The authors are quick to emphasise that the work is mostly still at a more conceptual stage, rather than providing a state-of-the-art algorithm, but the reviewers were generally perfectly happy with this given the contributions the paper does make; a view that I am very happy to support.